# The Molecular Basis and Clinical Consequences of Chronic Inflammation in Prostatic Diseases: Prostatitis, Benign Prostatic Hyperplasia, and Prostate Cancer

**DOI:** 10.3390/cancers15123110

**Published:** 2023-06-08

**Authors:** Saheed Oluwasina Oseni, Corey Naar, Mirjana Pavlović, Waseem Asghar, James X. Hartmann, Gregg B. Fields, Nwadiuto Esiobu, James Kumi-Diaka

**Affiliations:** 1Department of Biological Sciences, Florida Atlantic University, Boca Raton, FL 33431, USA; 2H. Lee Moffitt Cancer Center and Research Institute, Tampa, FL 33612, USA; 3Department of Computer and Electrical Engineering, Florida Atlantic University, Boca Raton, FL 33431, USA; 4Department of Chemistry & Biochemistry, and I-HEALTH, Florida Atlantic University, Boca Raton, FL 33431, USA

**Keywords:** chronic inflammation, prostatitis, benign prostate hyperplasia, prostate cancer, neuroendocrine differentiation, cancer stemness, castration resistance, inflammasomes

## Abstract

**Simple Summary:**

In this review paper, we explore the role of chronic inflammation in the pathogenesis of prostatic diseases, such as chronic prostatitis, benign prostatic hyperplasia (BPH), and prostate cancer. Chronic inflammation is a significant risk factor for these diseases, but the molecular mechanisms behind it are not yet fully understood. Previous attempts to investigate, link, and target inflammatory signaling molecules in men with these conditions have been clinically ambiguous and inconclusive. As a result, we proposed a paradigm shift in which less-explored molecules in the inflammatory signaling cascade are investigated as possible therapeutic targets for prostate diseases. In addition, we comprehensively discuss how aberrant signaling of these molecules may cause prostate cancer stemness, neuroendocrine differentiation, castration resistance, metabolic reprogramming, and immunosuppression. In conclusion, we forecast that targeting the interleukin-1 receptor-associated kinases (IRAKs) signaling pathway may provide a far more effective therapeutic or prophylactic strategy for the management of chronic inflammation-driven prostatic diseases.

**Abstract:**

Chronic inflammation is now recognized as one of the major risk factors and molecular hallmarks of chronic prostatitis, benign prostatic hyperplasia (BPH), and prostate tumorigenesis. However, the molecular mechanisms by which chronic inflammation signaling contributes to the pathogenesis of these prostate diseases are poorly understood. Previous efforts to therapeutically target the upstream (e.g., TLRs and IL1-Rs) and downstream (e.g., NF-κB subunits and cytokines) inflammatory signaling molecules in people with these conditions have been clinically ambiguous and unsatisfactory, hence fostering the recent paradigm shift towards unraveling and understanding the functional roles and clinical significance of the novel and relatively underexplored inflammatory molecules and pathways that could become potential therapeutic targets in managing prostatic diseases. In this review article, we exclusively discuss the causal and molecular drivers of prostatitis, BPH, and prostate tumorigenesis, as well as the potential impacts of microbiome dysbiosis and chronic inflammation in promoting prostate pathologies. We specifically focus on the importance of some of the underexplored druggable inflammatory molecules, by discussing how their aberrant signaling could promote prostate cancer (PCa) stemness, neuroendocrine differentiation, castration resistance, metabolic reprogramming, and immunosuppression. The potential contribution of the IL1R-TLR-IRAK-NF-κBs signaling molecules and NLR/inflammasomes in prostate pathologies, as well as the prospective benefits of selectively targeting the midstream molecules in the various inflammatory cascades, are also discussed. Though this review concentrates more on PCa, we envision that the information could be applied to other prostate diseases. In conclusion, we have underlined the molecular mechanisms and signaling pathways that may need to be targeted and/or further investigated to better understand the association between chronic inflammation and prostate diseases.

## 1. Introduction

Despite their different etiologies, most prostate pathologies or diseases have similar symptoms in men [1]. The most common forms of prostate pathologies are prostatitis, benign prostate hyperplasia/enlargement (BPH), and prostate cancer (PCa) [2].

Since the seminal discovery of inflammatory infiltrates (immune cells) in neoplastic tissues by Rudolf Virchow in 1863, many epidemiological, cellular, and molecular biology studies have been implemented to investigate and elucidate the association between inflammation and carcinogenesis [3]. Cases of prostatitis are often observed during the diagnosis of PCa, as chronic inflammation has been suggested as playing a critical role in the growth of cancerous cells. To better understand how aberrant inflammatory signals contribute to prostate pathologies, it is crucial to understand the molecular pathogenesis of aberrant inflammatory signaling in each prostate condition.

Normal inflammation, such as the healing of a wound or injured area in the body, is said to be “self-limiting” [4]. When the inflammatory response does not stop and seems to be triggered by other factors rather than bodily injury, the inflammation is referred to as chronic inflammation. Prolonged or recurring inflammation of this kind is often observed histologically in many cancers, including PCa. This type of inflammation can damage DNA over time and decrease cells’ ability to repair DNA, leading to uncontrolled cell growth, including malignant cells, and consequently resulting in the growth or progression of a tumor [5,6]. Multiple types of cancer have been linked to chronic inflammation. For example, colon cancer commonly occurs as a result of inflammatory bowel diseases such as Crohn’s disease, and stomach cancer can occur as a result of untreated *Helicobacter pylori* infection, which is a type of chronic gastric inflammation [7,8].

There are so many reasons why a better understanding of the role of chronic inflammation in the development, progression, and recurrence of prostatic diseases is important. One of these is the ability of chronic inflammation to remodel the extracellular matrix (ECM) of tissues and initiate the epithelial-mesenchymal transition (EMT) of prostatic cells. This occurs commonly during prostate carcinogenesis in which the prostate epithelial cells go through several biochemical changes, pressuring them to differentiate into aggressive and metastatic mesenchymal-like cells [9]. A single infection is enough to cause chronic inflammation, which is why cancer patients and immunocompromised individuals have chronic inflammatory co-morbidities [10]. The most common pathogens associated with bacterial prostatitis are *E. coli* and *Enterococcus* spp. However, the list is not limited to those two bacteria, as many other bacteria such as *Pseudomonas* spp., *Proteus mirabilis*, *Klebsiella* spp., and *Serratia* spp. have also been identified and isolated from the prostate during bacterial prostatitis [10,11].

## 2. Prostate Pathologies and Co-Morbidities

The prostate gland is a walnut-shaped organ located at the neck of the bladder, anterior to the rectum, with the male urethra running through it. The purpose of the prostate, together with sperm from the testicles and fluid from other glands, is to produce semen, which lubricates the sperm. The prostatic fluid is alkaline, nourishes the sperm cells and helps neutralize the acidity of the female reproductive tract during ejaculation [12]. Aside from fluid production, the muscles associated with the prostate gland also help make sure that the semen is pressed against the walls of the urethra and then expelled outwards during ejaculation. The human prostate gland is made up of three different histological zones known as the peripheral, transition, and central zones [10]. The peripheral zone is found on the prostate’s outermost portion and makes up about 70% of the tissue while the transition zone contains about 5% of the tissue in the prostate. The peripheral zone is the main site for the development of the majority of tumors in the prostate, while the transitional zone is relatively inconspicuous in younger men and found primarily in older men, in which it appears to be mostly enlarged due to BPH [13]. Tumor growth in the prostate is very common among men over the age of 40. More than 90% of men over the age of 80 are diagnosed with PCa and it has become so rampant in elderly men that it is considered to be a part of the aging process [14,15,16]. Other very common prostate pathologies of clinical significance include prostatitis and BPH. A study involving 5053 United States health officials found that 57.2% of men with prostatitis reported a history of BPH comorbidity whereas, in another study, 38.7% of the 7465 men who had BPH reported a history of prostatitis comorbidity [17].

## 3. Causal Inflammatory Drivers of Prostate Inflammation (Prostatitis)

Inflammation is an immune response against various factors, such as toxic compounds, damaged cell components, and pathogens, among others. The word inflammation was coined from “inflammo” which is a Latin word for “set ablaze/burn”, “inflame”, “kindle” or “set on fire” [18]. The National Library of Medicine describes inflammation as the immune system’s response to stimuli [18]. Hence, an infection, injury, cellular stress, or cellular damage can cause either acute, subacute, or chronic inflammation [19]. If the inflammation is controlled or acute, it helps the body repair itself and kills off pathogens, but if the inflammation is prolonged or not under control, it could damage DNA and cellular molecules and, at the same time, prevent the immune system from recognizing mutated or altered cells [19]. Recurring or prolonged inflammation, also known as chronic inflammation, can damage and repair cellular DNA, thus creating genetic instability within prostate cells, which attracts the production of many cytokines, chemokines, and growth factors [20].

Prostate inflammation, also known as prostatitis, is one of the most common pathologies of the prostate in men [21]. It is characterized as the inflammation or swelling of the prostate gland. Prostatitis is often presented as a comorbidity during the diagnosis of PCa and BPH. Prostatitis can be categorized into four subsets: Acute Bacterial Prostatitis (Category 1), Chronic Bacterial prostatitis (Category 2), Chronic Prostatitis or Chronic Pelvic Pain Syndrome (Category 3), and Asymptomatic Inflammatory Prostatitis (Category 4) [22]. Chronic Pelvic Pain Syndrome (CPPS) is characterized by symptoms including pain and lower urinary tract dysfunction but is often difficult to diagnose and treat. The actual etiology is still a question of research. However, the elimination of infectious agents as the source of prostate inflammation, in addition to a few known symptoms that are characteristic of CPPS, is best used to differentiate CPPS from microbial prostatitis during diagnosis [23]. Since the causal drivers of CPPS have not been fully defined, there is an urgent need for additional studies to decipher the molecular mechanisms underlying its pathogenesis, as well as for therapeutic strategies to effectively manage the condition in men [24].

In prostate biopsies showing lesions due to chronic inflammation, the risk of high-grade PCa was shown to be higher than those without these lesions [25]. In a study performed by the California Men’s Health Study (CMHS), a link was found between prostatitis and PCa, thus suggesting an increase in the risk of developing PCa in men with a history of chronic prostatitis [10]. Another study linked a history of clinical prostatitis to the significant increase in the odds of developing PCa in the general population [21]. As previously mentioned, many external factors promote the development of PCa; a particular intrinsic and extrinsic factor that is gradually restimulating the interest of many researchers is pro-tumorigenic inflammation developing in the prostate as a result of causal factors such as genomic instability, epigenetic changes, hormonal changes, infections or microbial dysbiosis, dietary factors, and other unknown factors from the environment [10].

Although unresolved, there have been attempts through diverse studies to explain the link between chronic inflammation (prostatitis) and PCa. Common risk factors for prostate inflammation are usually multiple and could be of pathogenic origins, such as bacteria, viral, or fungi infection, as well as of non-pathogenic/sterile origin, such as hormonal instability, autoimmunity, urine reflux, oxidative stress, among others [26,27]. Older men are especially known to be at a high risk of developing prostatitis and/or PCa. Both acute and chronic inflammation can be caused by pathogens responsible for sexually transmitted diseases and urinary tract infections [28]. Urinary tract infections caused by *Escherichia coli* (Uropathogenic *E. coli*, UPEC), and to a lesser degree, *Pseudomonas aeruginosa,* rank among the most common infectious diseases in adult men [29]. During infection, UPEC and other urinary tract infections or sexually transmitted disease-causing pathogens can reportedly alter or activate proinflammatory, pro-differentiation, proliferation, survival, antimicrobial, and various cell death (apoptotic) pathways in the host’s urogenital tissues, including the prostate gland, resulting in the development of preneoplastic lesions [30].

## 4. Causal Inflammatory Drivers of Benign Prostatic Hyperplasia/Enlargement (BPH/E)

Like all tumors, prostate growths are either benign (non-cancerous) or malignant (cancerous). When there is a benign growth in the prostate, this becomes what is known as benign prostatic hyperplasia (BPH). BPH is defined as the enlargement of the prostate in men [31,32]. As men age, the likelihood of BPH occurrence increases, with BPH affecting about half of men over 50 years of age, and nearly all men over the age of 80 [33]. Inflammatory features have been reported histologically in 30–43% of BPH samples [34]. The difference between prostatitis and BPH is that BPH does not have a known cause other than old age, while prostatitis is mainly caused by infection, possibly from bacteria or viruses [35]. BPH is frequently mistaken as PCa due to the very similar symptoms they share, although it is said that BPH does not increase the risk of individuals developing PCa [36]. Clinically diagnosed BPH causes bladder outlet obstruction (BOO), due to the lower urinary tract symptoms (LUTS) that come along with benign prostatic enlargement (BPE) [17,22]. The symptoms of BPH include a sudden urge to urinate due to incomplete bladder emptying, straining while urinating, nocturia, dribbling while urinating due to a weakened urinary stream, pain while urinating, and blood within the urine. The mechanism behind the pathology of BPH and the symptoms observed are based on the degree of prostate enlargement. For instance, the capsule surrounding the prostate prevents it from expanding radially, thus resulting in urethral compression, leading to difficulty in urination [36]. Prostate tissue damage and chronic tissue healing have been suggested as contributing either directly or indirectly to the development of BPH nodules [37].

Through Roehrborn and colleagues’ research efforts, the symptoms associated with BPH have been demonstrated to vary in each patient. Some men remain unbothered by symptoms and will not seek medical attention, while other males with more intense symptoms will seek medical help to alleviate their symptoms. In addition, Roehrborn and colleagues found that Lower Urinary Tract Symptoms (LUTS) associated with BPH can be classified into two categories: Obstructive (weak flow, straining, and hesitancy) and Irritative (nocturia and painful urination) [38]. Dhingra and colleagues showed that 5α-reductase inhibitors, which catalyze the formation of dihydrotestosterone (DHT) from testosterone in specific tissues, including the prostate gland, contribute to the decrease in the levels of DHT in the prostate. Additionally, α1-AR antagonists decrease LUTS and increase urinary flow rates in men with symptomatic BPH. For example, in a study to investigate the effects of ibuprofen and doxazosin on BPH, both drugs were reported to significantly induce apoptosis and decrease the viability and proliferation of prostate cell lines, thus clinically relieving the symptoms of BPH [17,22].

Despite its high prevalence in older men, not many studies have been carried out at the molecular level to understand the pathogenesis of BPH. This is in part because BPH is assumed to be a lesser life-threatening illness compared to PCa. However, epidemiological reports have shown that BPH could also severely impact the quality of life of affected men, just like PCa [37,39]. To outline the genomic and mutational landscape of BPH, Liu and colleagues integrated both transcriptional and methylation analyses, in which they identified and validated two BPH subgroups with distinct clinical features and signaling pathways in two independent cohorts [40]. In their study, they found that mTOR inhibitors have the potential to target a specific BPH subtype, by demonstrating that men exposed to mTOR inhibitors show a significant decrease in prostate size. TRAF6 signaling has also been shown to promote the proliferation of BPH stromal cells via Akt/mTOR signaling [41]. Interestingly, stromal cells of the prostate contribute to inflammatory reactions in the transition zone of BPH tissues via TRAF6 signaling [42]. Hence, targeting aberrant signaling of TRAF6 could be a promising therapeutic strategy for BPH treatment.

Elevated expression of pro-inflammatory cytokines has been reported in BPH tissues by multiple studies [36,42,43]. However, little is known about the clinical significance of many of these pro-inflammatory signaling molecules in BPH pathogenesis. Recurrent activation of the IL-1/TLR signaling pathway in the prostate cells may play a significant role in inducing immune reactions mediated by innate and adaptive immune cells in BPH tissues. For instance, BPH cells have been demonstrated to express almost all of the TLRs. The activation of some of these TLRs was shown to increase the production of cytokines such as CXCL8/IL-8, CXCL10, and IL-6 in BPH cells [35]. Co-stimulatory molecules such as class I and class II MHC have also been observed to be highly expressed and activated in BPH tissues, thus leading to the enrichment of alloreactive CD4+ cells and upregulation of IL-12/IL-23p40 and IL-12p75 by BPH cells [44]. The enriched alloreactive CD4+ cells secrete IFN-gamma and IL-17 [45]. Proinflammatory cytokines and chemokines can recruit lymphomononuclear cells by acting as antigen-presenting cells (APCs). Upregulation of IL-6, IL-8, and CXCL10 by BPH cells creates a positive feedback loop that can amplify inflammation [44,45]. In another study, IL-8, CXCR1, CXCR2, and CXCR7 were found to be 5 to 25-fold elevated in BPH tissues relative to normal prostate tissues [46]. Furthermore, overexpression of IL-8 was shown to induce autocrine/paracrine proliferation of BPH cells, indicating the growth-promoting potential of this chemokine in BPH pathogenesis. Elocalcitol, a vitamin D analog and vitamin D receptor (VDR) agonist, has been shown to inhibit IL-8 production and intraprostatic cell infiltrate in a mouse model and in BPH patients [35].

JM-27, also known as prostate-associated gene protein 4 (PAGE4). is an androgen-regulated protein predominantly expressed in the prostate that is highly upregulated in symptomatic BPH and has been suggested to be involved in prostatic growth regulation [47]. Cytokines such as IL-6 and IL-17 were found to propagate chronic immune response in BPH patients by inducing fibromuscular growth via induction of COX-2 expression [48]. Downregulation of anti-inflammatory molecules such as macrophage inhibitory cytokine-1 (MIC-1) has been reported in symptomatic BPH tissues [49].

Put together, there is a great deal of missing information on the pathogenesis and treatment options for BPH. More studies are needed to elucidate the inflammatory and immunoregulatory components of BPH pathogenesis, at the cellular and molecular levels. Since BPH is very common among men as they get older, better, and more effective treatment options are needed for BPH. The association between BPH prevalence and aging, chronic prostatitis, or PCa requires further clarity.

## 5. Causal Inflammatory Drivers of Prostate Carcinogenesis

PCa is the most widely diagnosed and the second largest cause of cancer deaths in men residing in the United States. In 2023, it is estimated that there will be about 28,830 new cases of PCa, and about 34,700 men are likely to die from the disease [50]. Many risk factors have been associated with increased incidence and progression of PCa. This includes age (>40 years), race (more aggressive in blacks than other races), smoking, hormonal imbalance, and lifestyle, among others [51]. Genetic predisposition and mutation have also been reported to be responsible for the development of sporadic and familial PCa. Men who have immediate families with PCa are about 50% more likely to develop this disease [51,52]. Lifestyle modifications, including smoking cessation, exercise, and weight control are preventative steps an individual can take to decrease the risk of PCa occurrence, but early screening for PCa is one of the most important steps to prevent the aggressive and advanced form of the disease [53].

Prostate tumors are known to be heterogeneous with the ability to develop different cancer phenotypes in collaboration with stromal or tumor-associated immune cells (Figure 1). Tumor progression is characterized by phenotypic and genomic changes in tumor cells that lead to acquired aggressive behaviors, such as increased growth, metastasis, and invasiveness [54]. Following a failed response to therapy, the old or new tumor sites become repopulated by aggressive PCa phenotypes of different genomic backgrounds, such as castration-resistant PCa (CRPCs), stem PCa (PCSCs), and neuroendocrine PCa (NEPCs) [55,56]. This heterogeneous nature of prostate tumors favors cancer progression, which is a major challenge to treatment and the duration of remission in PCa survivors. Therefore, understanding the underlying mechanisms through which prostate tumors gain the ability to clinically progress, metastasize, or repopulate is critical for the development of novel therapeutic strategies to counter these phenomena.

As highlighted in the sections above, there are notable pieces of evidence to suggest that chronic inflammation may be involved in PCa initiation and/or progression, but support for this is still contentious [30]. Chronic inflammation is associated with higher rates of cellular mutations and genetic alterations, which could drive tumorigenesis. According to epidemiological data, chronic infection, and inflammation are associated with over 25% of all cancers [57]. During chronic inflammation, the pro-inflammatory and oncogenic genes, through prolonged activation of several inflammatory signaling pathways, growth proteins, and cellular messengers, continue to fire non-stop [58]. The body’s cells become more vulnerable to genomic instability and mutation under these conditions, leading to a microenvironment that is favorable for carcinogenesis [20].

Several pathogenic and non-pathogenic microbes have been isolated from the urine and prostate tissues of PCa patients using different microbiological and molecular techniques (Figure 2). Urogenital infections have been suggested as contributing to the pathogenesis of prostatic diseases, either directly or indirectly [59]. Chronic inflammation could also be driven by non-pathogenic sources of inflammation as a result of genetic or epigenetic changes in the prostate epithelial cells. Moreover, many inflammatory genes or proteins are often found to be overly or aberrantly expressed in aggressive or recurring PCa patients [60,61]. Finally, one of the biggest challenges encountered by oncologists when treating late-stage PCa is the appearance of drug-resistant phenotypes within the tumor microenvironment. Recent reports have begun highlighting the potential clinical significance of chronic inflammation in influencing PCa heterogeneity and increased enrichment and clonal growth of these aggressive PCa phenotypes [62].

Chronic inflammation can initiate the development and progression of cancer through its ability to damage DNA, thus promoting somatic mutations and structural variations in cells. Structural genetic alterations, such as somatic mutations and copy number variations, account for increased PCa risk among men of diverse backgrounds. For instance, inherited mutations of p53, PTEN, HOXB13, BRCA1, and BRCA2 genes have been associated with increased PCa risk and account for most cases of hereditary/familiar PCa [63,64]. Genes such as SPOP, ERG, and PTEN are recurrently mutated in primary prostate tumors. Somatic mutations in genes such as AR, p53, PTEN, ERK, and RAS have been associated with sporadic development and progression of PCa. Combined BRCA1/2 and ATM mutations have also been reported to be significantly higher in men with lethal PCa compared to those with localized PCa [64]. However, several other genes, including inflammatory genes that may be important to PCa development and progression, remain unexplored or uncharacterized. Additional genetic, epigenetic, and functional studies are required to better understand the impacts of the characterized and uncharacterized genes on PCa progression, as well as how to better improve the therapeutic potential of targeting these genes. Interestingly, genomic studies in different cancers, including PCa, have provided evidence that deregulation, as well as single nucleotide polymorphisms (SNPs) in genes associated with inflammatory pathways, may impact the risk of disease initiation, progression, and severity [65]. For example, SNPs in TLRs, IL-1β, IRAK1, and IRAK4 have been linked with an increased risk of PCa development and recurrence [66,67,68,69].

A five-year longitudinal study of prostate biopsies from suspected PCa patients shows that chronic inflammation may play a role in the development of PCa in approximately 20% of patients, determined by the presence of inflammatory cell infiltrates in tissues, and was found to be significantly correlated with the presence of areas of proliferative inflammatory atrophy (PIA) in the prostate tissues [9]. In the same study, PIA was found histologically in approximately 40% of examined PCa tissues, whereas other studies have been less convincing, reporting little or no association between chronic inflammation, PIA, and prostate adenocarcinoma (PRAD). There has also been recent interest in understanding the potential role of asymptomatic prostatic inflammation caused by infectious microorganisms or intracellular and extracellular sterile stimuli on PCa development and progression in vitro and mouse models [61,70].

Perhaps the best evidence of the importance of chronic inflammation signaling in PCa development was an epidemiological study that linked prolonged intake or administration of Aspirin, a non-steroidal anti-inflammatory drug (NSAID), to a lowered risk of developing PCa [71,72]. Low-grade or asymptomatic chronic inflammation may have been caused as a result of prolonged or continuous exposure to these risk factors at a moderate level. However, due to the position of the prostate gland, it is very difficult to detect such low-grade or asymptomatic chronic inflammation in indolent PCa patients. Hence, it is a potential candidate driver of prostate stemness, neuroendocrine differentiation, and castration resistance [73,74,75]. Other likely causes of high or low-grade inflammation and hyperactivity of the immune system include long-term exposure to the mild forms of the same stimuli, hyperactivation of the danger signals, inability to eliminate danger stimuli, uncontrolled activation of danger sensors, and age-related weakness in the regulation of the inflammatory and immune signaling mediators [76].

## 6. Role of The Human Microbiome in Tumor Development and Progression

The role of the human microbiome (or microbiota) in health is emerging and gradually gaining attention in the scientific world for its impact on homeostasis, metabolism, inflammation, diseases, immunity, hematopoiesis, neurological and cognitive functions [72,77,78]. The human microbiome is the accumulation of microbial genes and genomes in the body and the summation of all microbiotas that reside on or within human tissues, including the biofluids, in proximity to the anatomic sites in which they reside [79]. The nature of the microbiota in the various tissues and organs in the body has been associated with the health status of the patient and could be categorized as either a good microbiome or a bad microbiome (Figure 3 and Figure 4). Organs with a good microbiome have a microbial ecosystem that exhibits symbiosis to prevent the overgrowth of pathogenic organisms that could cause disease conditions, while the opposite occurs for organs with a bad microbiome.

Microbiota is the term used to describe a collection of microorganisms—viruses, bacteria, archaea, fungi, bacteriophage, and protozoans—specific to a particular body site or habitat. The composition of microbiota in each human tissue varies due to several factors, including genetic factors, colonization during conception, the delivery method at birth, host lifestyle, exposure to antibiotics, medications, dietary intake, cultural beliefs, and diseases [80]. Microbial dysbiosis occurs due to imbalances in the dynamics of the microbiota of tissues caused by internal and external factors, such as exposure to chemical agents, changes in diets or lifestyle, smoking, aging, and hormonal imbalances, among others. The dynamics of the microbiota community of each tissue fluctuate according to environmental changes going on in the specific tissue in which the microorganisms reside [81]. Massari et al observed a sex-related disparity in the urinary microbiota. They observed a high abundance of *Lactobacillus* and *Gardnerella* in the female, while *Corynebacterium*, *Staphylococcus*, and *Streptococcus* are the most abundant urinary microbiota in the male [81].

Evidence from a plethora of studies indicates that pathogenic microbes are responsible for over 20% of cancer cases [82,83]. In some cases, the microbiota was shown to influence cancer development by modulating inflammation signals and inducing genomic instability of host cells [84,85]. In their studies, both research groups provided seminal evidence to show that transplanted gut microbiota can influence cancer development and that the microbiota of mice could be manipulated to support the development of tumors through microbial dysbiosis. In the study by Zackular et al, germ-free mice were conventionalized with gut microbiota from tumor-bearing donor mice, and the effects were compared to gut microbiota from non-tumor healthy mice. The microbiota-conventionalized mice showed a 2-fold increase in tumor burden/development relative to those who received the transplant of gut microbiota from healthy mice donors.

### 6.1. Role of The Prostate Microbiota in Prostate Tumorigenesis

Given that the prostate gland is in proximity to the bladder and urinary tract, many studies have proposed that microbes and/or inflammatory stimuli from the urinary tract could promote tumorigenesis in the prostate gland [81]. One of the major causes of prostate inflammation is pathogenic bacteria. Bacterial prostatitis is often associated with infection by species of Enbacteriaceae, including *E. coli.* Other uro-pathogens have also been identified, isolated, and characterized using several conventional and molecular techniques (Figure 2). There have been studies aimed at linking chronic inflammation and urinary microbial dysbiosis to PCa, with mixed results [86]. Using ultradeep pyrosequencing, Cavarretta et al found that the microbial abundance of some species of bacteria is tumor-specific and may vary between the different architectural zones of the organ housing the tumor [87]. In their study, the relative microbial abundance for tumor, non-tumor, and peri-tumor prostate tissue samples collected from 16 Caucasian PCa patients after radical prostatectomy was evaluated and found to differ between the transitional and peripheral zones of the prostate [88].

The dominant phylum identified was the gram-positive *Actinobacteria.* while the most abundant genera were *Propionibacterium* followed by *Firmicutes* and *Proteobacteria* in all three different prostate tissue sample types analyzed. However, *Staphylococcus* spp. was common in the tumor and peri-tumor tissues compared to the non-tumor tissue samples [88]. Furthermore, men with biopsy-positive PCa have a higher proportion of the bacteria often associated with urogenital infections, such as STDs and UTIs, compared to biopsy-negative subjects [60]. A high abundance of bacteria, such as *Varibaculum cambriense*, *Streptococcus anginosus*, *Anaerococcus obesiensis*, *Actinobaculum schaalii*, *Anaerococcus lactolyticus*, and *Propionimicrobium lymphophilum,* in prostate tissue biopsies have been reported [60]. *Mycoplasma genitalium* was reported to be common in PCa patients relative to BPH patients [89]. In patients with PCa, bacteria associated with carbohydrate metabolism pathways were also in abundance whereas bacteria associated with folate, riboflavin, and biotin were less abundant [90].

The rectal microbiome profiling of rectal swabs from men before transrectal biopsy reveals a significant increase in the microbial abundance of pro-inflammatory *Bacteroides* and *Streptococcus* species in those diagnosed with PCa compared to those without PCa. In fecal samples of healthy male volunteers versus men with localized or biochemically metastatic PCa, there was a higher abundance of *Akkermansia muciniphila* and Ruminococcaceae in the latter. Significant compositional differences have been reported in the microbiota of men being treated with oral androgen receptor axis-targeted therapies, such as bicalutamide, enzalutamide, and abiraterone acetate [91]. In an earlier article from the same laboratory, a single bacterial infection was shown to have the potential to induce long-term chronic inflammation in the prostate of mice and that this infection-induced chronic inflammation could persist for months and years, post-infection. In PCa patients taking androgen deprivation therapy (ADT), a higher relative abundance of *Akkermansia muciniphila* and Ruminococcaceae also known as Ruminoclostridium has also been described [92,93].

*Mycoplasma hyorhinis* can metabolize gemcitabine, a drug prescribed for PCa patients, into an inactive metabolite, therefore decreasing the efficacy of the drug [93]. ADT has also been shown to have a significant impact on intestinal microflora dynamics, thus inducing abdominal obesity. Castrated mice with androgen deficiency were found to have increased *Firmicutes/Bacteroidetes* ratio and *Lactobacillus species* in their feces compared to the control [94]. *Bifidobacterium* spp. was found to be inversely associated with obesity and not affected by castration or decreased androgen. Since ADT is the go-to therapeutic strategy for early PCa treatment and has been reported to promote the development of obesity, ADT may contribute to prostate microbiome dysbiosis, either directly or indirectly [95]. However, more studies will be needed to understand how ADT influences microbial abundance and dysbiosis in the prostate tumor microenvironment.

### 6.2. Mechanisms of Microbial Dysbiosis in Prostate Diseases

Microbial dysbiosis has been implicated in several disease conditions and has been reported to be responsible for mixed patient-based responses to different therapeutic strategies, including radiotherapy, chemotherapy, phytotherapy, and immunotherapy [96]. A plethora of mechanisms have been suggested to cause microbial dysbiosis (bad microbiome) which could lead to PCa development and progression (Figure 4). One of the notable hypotheses is that microbial dysbiosis encourages the abundance of pathogenic microbes, which aberrantly stimulates the TLR/IL-1R/NF-κB signaling pathway, resulting in chronic inflammation and cytokine storming. This leads to overproduction and overexpression of oncoproteins, oncogenic metabolites, and oxidative stress products (ROS and RNS) causing genetic instability, mutations, and angiogenesis that predispose to cancer development and survival.

In one example, chronic inflammation could promote focal prostatic glandular atypia [97]. The authors observed that mice infected with *E. coli* developed both acute and chronic bacterial prostatitis, while the control mice receiving phosphate-buffered saline (PBS) had no prostate infections or inflammation. The outcome of their study suggested that the dose, infectivity, and duration of infection with the pathogen determine the degree of chronic inflammation-driven carcinogenesis. For instance, in their study, mice infected for 5 days showed signs of acute inflammation with infiltration by neutrophils and epithelial necrotic debris in the prostatic glandular lumen, while mice infected for 12 weeks showed evidence of chronic inflammation with dense inflammatory infiltrates in the stroma. The prostatic epithelium at this stage showed varying degrees of atypical hyperplasia with increased epithelial cell layers and cytological atypia. However, at 26 weeks of infection, a pronounced and reactive dysplasia was observed in the mice. The lesions observed in the prostate organs of the mice resemble both prostatic intraepithelial neoplasia (PIN) and high-grade dysplasia.

The prostatic glands exhibiting reactive dysplasia showed high oxidative DNA damage, increased epithelial proliferation, and a decrease in the androgen receptor and *GSTP1*, *p27Kip1*, and *PTEN* expression relative to control prostate glands [97]. This finding is one of many studies that suggest a potential connection between chronic inflammation and prostatic neoplasia; this study particularly showed that chronic inflammation promotes focal prostatic glandular atypia.

Overall, these findings provide pieces of evidence to support the proposition that the molecular and microbial dynamics of the prostate, as well as the gut, influence prostate tumor development, progression, and response to therapy. While it is easier to conclude that changes in the microbial abundance of prostate microbiota contribute to cancer progression, there is the possibility that their relationship may be symbiotic, in which the tumor cells undergo metabolic reprogramming to induce changes in their microenvironment in order to encourage the growth of tumor-promoting microbes. An example is an increased abundance of Lactobacillales (lactic-acid bacteria (LAB) or lactate-utilizing microbes), including bacteria of the genus *Lactobacillus, Aerococcus, Enterococcus, Leuconostoc, Streptococcus, Carnobacterium, Tetragenococcus, Vagococcus, Sporolactobacillus, Weissella, Oenococcus,* and *Pediococcus* around tumor sites [98]. These microbes have increased tolerance to the low pH or acidic nature of the tumor microenvironment and could outlive other non-LAB microbes in the tumor environment. The presence of the microbes and the lactate-surplus environment may provide signaling mechanisms that favor the production of more tumor-promoting factors. While this is still a hypothesis and highly debatable, other studies have suggested that the presence of these LAB microbes may have anti-tumor effects in colorectal cancer [99,100]. More studies will be needed to confirm this hypothesis, especially in the case of prostate tumors.

## 7. Role of Sterile Inflammation in Prostate Carcinogenesis

Relief from prostatitis after intake of antibiotics is not always guaranteed. This is because prostatitis is not dependent solely on the persistence of pathogenic bacteria. Several other factors and stimuli, such as hormonal alterations from androgen or estrogen imbalances, can exacerbate the inflammatory lesions in the prostate. Urine reflux or exposure to certain chemicals could also cause architectural alterations, as well as physical trauma consequent to *Corpora amylacea*. Carcinogens in diets may also reach the prostate gland through the vascular system and cause DNA damage leading to sterile inflammation [81].

Elevated levels of estrogen have been reported in patients with PCa compared to healthy controls [10,101]. Estrogen promotes carcinogenesis by activating polycyclic aromatic hydrocarbons (PAHs), which have been implicated in the formation of carcinogen metabolites, diol-epoxies, and radical cations. Plottel et al identified and proposed functional estro-biome and enteric bacterial gene signatures that can metabolize estrogen by conjugation and deconjugation [102]. Trauma caused by *Corpora amylacea* or urine reflux could also induce chronic inflammation and abnormal epithelial cell regeneration. Increased production of reactive oxygen species (ROS) and reactive nitrogen species (RNS) have been observed in the inflammatory microenvironment, which results in the development of inflammatory lesions in certain areas of the prostate tissue known as proliferative inflammatory atrophy (PIA). PIA is a known precursor to low-grade and high-grade prostate intraepithelial neoplasia (PIN) leading to the formation of prostate adenocarcinoma (PRAD) [103].

## 8. Impact of Racial Disparities in Chronic Inflammation-Driven Prostate Diseases

It is common knowledge that cancer health disparities exist among populations of diverse ethnic/racial backgrounds. The race/ethnic background of an individual has been shown to play an important role in cancer aggressiveness and lethality. Genetic predisposition due to varying genetic alterations or mutational landscape is one of the most significant risk factors known to contribute to high PCa aggressiveness and deaths among African-American men. In the United States, African-American men have the highest incidence of PCa and are twice as likely to die from PCa compared to other races. The risk of PCa diagnosis was reported to be 2.2 times (95% CI: 1.48–3.35, *p <* 0.001) greater in African-American than Caucasian men in analyses adjusting for serum PSA level [104]. African-Americans with BPH have a much greater risk of developing PCa than similar Caucasian men [105].

Interestingly, a few studies have reported a higher prevalence of chronic inflammation in non-tumor prostate biopsy specimens from African-American men compared to European-American men [28]. Other studies have identified several PCa-associated genes as differentially expressed between African-American and European-American men. Thus, suggesting a higher pre-malignant chronic inflammation may be a contributor to the aggressive nature of PCa in African-American men. Stromal cells in the tumor microenvironment of African-American men may also be contributing to PCa progression by increasing the expression levels of some pro-inflammatory molecules compared with European-American men [106]. SNPs of some inflammatory mediators, such as TLR3, TLR6, TOLLIP, IRAK1, IRAK4, and IRF3, have been reported in PCa men of African, European, and Asian ancestry [67,68,69].

Despite emerging knowledge and studies on this topic, there is still much unknown about the relationship between race and chronic inflammation in cancer. Little is known about the molecular mechanisms by which chronic inflammation contributes to health disparities in different race/ethnic groups. Therefore, additional cancer health disparity studies are needed for a better understanding of the inflammatory and immunologic landscape of the prostate in diverse populations. The ability to identify the molecular drivers as well as the impacts of chronic inflammation on the severity of prostate pathologies in men of different racial/ethnic backgrounds at the cellular, molecular, and clinical levels will provide strategic options for the prevention and treatment of these diseases. Future studies should also be focused on evaluating whether genomic, transcriptomic, proteomic, epigenomic, metabolomic, and micro-biomic alterations in the prostate of African American men predispose them to a higher prevalence of chronic prostatitis, as well as prostate tumor progression, metastasis, and recurrence.

Race/ethnicity has been associated with the risk of developing BPH. The degree of androgen receptor signaling of prostatic cells may contribute to differences in the prevalence of BPH between men of diverse racial backgrounds [107]. The risk of BPH is more than 40% higher among black and Hispanic men when compared with Caucasian men [105]. The prostate volume was also reported to be greater in Caucasian men than in Asian men and found to be consistent with increasing age. PSA level, which is a commonly used biomarker for diagnosing prostatitis and PCa, has been reported to increase rapidly over time in African-American men relative to Caucasian-American men [104], thus suggesting genetic that epigenetic, and microbiome factors may be important drivers of racial/ethnic differences in serum PSA levels among men.

Disparities in the microbiome component between normal tissues and tumor tissue samples have been reported and found to vary based on race, age, gender, geographical location, diet intake, and other demographic factors [108]. A study on the gut microbiome and the expression levels of gut microbiome-associated genes of individuals of Asian, European, and American origin identified race-related disparity as influencing the microbiome composition and diversity of the study participants. The study also identified 400 microbiome-associated genes shown to distinguish the gut microbiome between the different racial populations [108]. From this study, we can suggest that a better understanding of the impact of the race–microbiome relationship on chronic inflammation-associated prostatic conditions is urgently needed.

## 9. Mechanisms of Oncogenic Inflammatory Signal Transduction

Chronic inflammation is regarded as the seventh hallmark of carcinogenesis and as a possible trigger for tumor initiation and progression at the cellular and molecular levels. However, the molecular mechanisms linking chronic inflammation signaling and PCa progression are not well understood [4]. Unfortunately, due to the complexities of the inflammatory signaling pathways, many previous clinical studies investigating the therapeutic potentials of upstream (TLRs and IL-1Rs) and downstream (NF-κB) inflammatory signaling proteins have been clinically unsuccessful and have produced inconsistent results [109]. The majority of these studies have also neglected the tumor-promoting potentials of midstream proteins, which act as inflammatory regulators and adapters for upstream and downstream mediators of inflammation [110,111]. Hence, the overarching goal of our research group is to investigate and elucidate the oncogenic role of the less-studied midstream regulatory and adaptor proteins in the inflammatory cascade, particularly the role of interleukin-1 receptor-associated kinases (IRAKs) in prostate tumor progression, as well as to gain insight into their diagnostic, prognostic, and therapeutic potentials in chronic inflammation-driven PCa [112,113,114]. Pathogen-associated molecular patterns (PAMPs) and damage-associated molecular patterns (DAMPs) are alarming molecules involved in the initiation of the inflammatory cascade expressed by pathogens (microbes) and damaged/dying cells or their intracellular components, respectively [115]. There are about 50 pattern recognition patterns (PRRs) in mammals, divided into two classes: the Membrane-bound Recognition Receptors, such as Toll-like Receptors (TLRs) and C-type Lectin Receptors (CLRs), and, secondly, the Cytosolic Pattern Recognition Receptors (CPRRs), which include RIG-1-like Receptors (RLRs), Nucleotide-binding Oligomerization Domain (NOD)-like receptors (NLRs), Absent in Melanoma 2 (AIM2)-like Receptors (ALRs), and other Nucleic Acid-sensing Receptors like cGAS and STING [116,117,118]. Upon sensing various DAMPs or PAMPs, pattern recognition receptors (PRRs), such as toll-like receptors (TLR1-10), become activated and, along with interleukin 1 receptors or cytokines (IL1-Rs), initiate a series of downstream signaling events via their adaptors to drive cellular responses, aided by the activation of transcriptional factors. This culminates in the production of proinflammatory cytokines, such as TNF-α, IL-1, IL-6, IL-18, and IL-12, and chemokines, such as CCL-5, CXCL-8, and MCP-1, type-1 interferons (IFN), among others, to orchestrate inflammation, cell survival, migration, and immune response [119,120].

Specifically, the activation of Myeloid differentiation primary response 88 (MyD88) adaptor-dependent pathway by TLRs (except TLR3) or IL-1Rs leads to recruitment and interaction with IRAK4 and IRAK2 to form a complex called Myddosome. IRAK4 later binds and phosphorylates IRAK1 through its death domain (DD). The phosphorylated IRAK1 dissociates from the Myddosome and further auto-phosphorylates through its kinase activity to activate Lys63-linked polyubiquitination of TRAF6, which forms a complex with TGF-β-activated kinase 1 (TAK1), TAK1-binding protein 1 (TAB1), TAB2, and TAB3. The activation of this complex triggers the recruitment and activation of downstream transcription factors, such as early-phase NF-κB, and AP-1, followed by the secretion of pro-inflammatory cytokines and mediators, such as IL-1α/β, IL-6, IL-12, and TNF [119,121,122]. The pro-inflammatory cytokines induce and maintain inflammation, which may become a problem (chronic) when continuously induced for a prolonged period. Other important downstream partners include type-1 Interferons and cell pro-survival genes, such as MAPK, PI3K, JNK, p38, and ERK isoforms [123].

Overall, the intracellular dysregulation of IL1/TLR/IRAK signaling within tumor cells or infiltrating immune cells in the tumor microenvironment may contribute to chronic inflammation-driven cancer survival, growth, invasion, immune evasion, and chemoresistance [124,125,126,127].

### 9.1. Pathogen-Associated Molecular and Damage-Associated Molecular Patterns

Inflammatory conditions are triggered by two mechanisms: either exposure to exogenous ligands from pathogenic microbes during an infection, known as PAMPs, or exposure to intracellular or extracellular stress-induced molecules or tissue damage components, known as DAMPs [115,128,129]. Both PAMPs and DAMPs are sensed by danger sensors called PRRs on the cell surface of the host or in endosomes [116]. PRRs are expressed by both innate immune cells (neutrophils, eosinophils, basophils, macrophages, dendritic cells, and monocytes), as well as some epithelial cells of organs, such as the prostate, brain, tongue, stomach, small and large intestines, skin, and breast [130,131]. High expression of PRRs has also been observed in malignant tumors from these organs [117].

### 9.2. Pattern Recognition Receptors: Toll-like Receptors and Their Adaptors

Toll-like receptors (TLRs) are the most widely studied pattern recognition receptors (PRRs) in humans [119,132,133]. Currently, there are about 10 (TLR1–10) and 13 (TLR1–13) TLRs in humans and mice, respectively, and all are capable of being activated by different PAMPs and DAMPs [117]. TLRs on cell membranes act as important sensors of foreign microbial components, as well as products of damaged or inflamed self-tissues [134]. Upon sensing these molecules, TLRs initiate a series of downstream signaling events that drive cellular responses aided by the production of proinflammatory cytokines, type-1 interferons (IFN), co-stimulatory molecules, and chemokines [118]. TLR2 has been shown to heterodimerize with TLR1 or TLR6, while TLR10 can also heterodimerize with TLR1 or TLR2. PAMPs (exogenous antigens) and DAMPs (endogenous antigens) activate diverse TLRs and subsequently initiate the downstream signaling pathways [135] (Figure 4).

Four TLR adaptor proteins have been characterized: MyD88 (myeloid differentiation factor 88), TRIF (Toll-receptor-associated activator of interferon), MAL/TIRAP (MyD88-adaptor-like/TIR-associated protein), and TRAM (Toll-receptor-associated molecule). These adaptors are known to participate in signal transduction, activation, and regulation of protein kinases, as well as transcription factors downstream of the inflammatory pathways, via their TIR (Toll/interleukin-1 receptor (IL-1R) homologous region) domains [136].

### 9.3. Mechanisms of Activation of PRRs by PAMPs

As seen in Figure 5 and Figure 6, TLRs can be activated by diverse molecules on the surface of microorganisms, known as PAMPs. TLR1 can be activated by triacylated lipoprotein from bacteria, and atypical bacteria lipopolysaccharides (LPS), whereas TLR2 can recognize peptidoglycans (PDG) from Gram-positive bacteria, fungal (yeast) and bacterial zymosan, lipoteichoic acid (LTA) from Gram-positive bacterial, a glycoprotein from *Treponema maltophilium*, phospho-lipomannan from *Candida albicans*, phenol-soluble modulin from *Staphylococcus aureus*, GPI anchor from *Trypanosoma cruzi* and *Plasmodium falciparum* and a synthetic bacterial lipopeptide known as PAM_3_CSK_4_ [137]. TLR3 can recognize viral double-stranded RNA (dsRNA) and synthetic dsRNA Poly(I:C) or Poly(A:U), while TLR4 is activated by lipopolysaccharides (LPS) from Gram-negative enterobacteria, mannan from *Saccharomyces cerevisiae* and *Candida albicans*, and glucuronoxylomannan from *Cryptococcus neoformans* [138]. One of the most common gram-negative bacteria in the prostate or urinary tract is the Uropathogenic strains of *E. coli,* known as UPEC strains [81,139]. UPECs are a genetically and phylogenetically diverse subgroup of extraintestinal pathogenic *E. coli* (ExPEC) strains that infect organs outside the gastrointestinal tracts, including the urinary tract, lungs, and bloodstream. TLR5 can sense flagellin derived from flagellated bacteria while TLR6 (CD286) is activated by triacylated lipopeptides of bacteria and diacetylated lipopeptides from *Mycoplasma* sp. Within the endosome, TLR7 can sense both viral and bacterial single-stranded RNA (ssRNA), while TLR8 can sense both viral and bacterial single-stranded RNA (ssRNA), phagocytized bacterial RNA, and synthetic imidazo-quinoline derivatives, such as Imidazole, Gardiquimod, and Resiquimod [140]. TLR9 is also located in the endosome and can sense unmethylated CpG islands of bacterial and viral DNA, hemozoin from *Plasmodium falciparum*; TLR10 recognizes triacylated lipopeptides from bacteria; TLR11 can recognize profilin from *Toxoplasma gondii* and Uro-pathogenic bacteria; TLR12 can recognize profilin from *Toxoplasma gondii*; TLR13 can recognize bacterial ribosomal RNA (rRNA); and newly discovered TLR14 can be activated by an uncharacterized polypeptide that acts as an inhibitor of the bacteria cell wall [141]. 

### 9.4. Structural Architecture and Oncogenic Functions of Bacteria Lipopolysaccharides and Lipoteichoic Acids

The bacteria lipopolysaccharide (LPS) is the most important structural and immunodominant component of the outer cell membrane complex of Gram-negative microorganisms in terms of the bacteria virulence/pathogenicity and its ability to serve as an alarmin [138]. Structurally, the LPS, a low molecular weight carbohydrate with a molecular mass of 10–20 kDa, constitutes approximately 75% of the Gram-negative bacteria cell membrane and 5–10% of the organism’s dry mass. It consists of three components: Lipid A, responsible for anchoring the LPS in the membrane; core oligosaccharides; and the "O" antigen, a polymer of repeating oligosaccharides that varies between species and contributes to serological specificity (Figure 7A). Lipid A is known to be responsible for the endotoxic effects associated with bacterial LPS. Pathogenic and non-pathogenic bacteria organisms can release their LPS molecules, which bind to cell membrane receptors, particularly to TLRs, and stimulate PAMPs. LPS can stimulate an inflammatory cascade in both human epithelial and immune cells through the activation of TLR4, which is located on the cell membrane of these cells [137]. TLR4 is one of the most widely studied members of the TLR protein family; activation of TLR4 by LPS from Gram-negative bacteria stimulates downstream inflammatory cascades that result in proinflammatory signaling transduction and cytokine production. Hence, LPS is known as a TLR4 agonist or ligand [142]. Importantly, recurrent exposure to LPS from bacteria has been shown to enhance PCa metastasis, potentially through activation of NF-κB signaling pathways and increased chemoresistance to dexamethasone administration [143].

LPS can also induce overexpression of CCL2 (C-C motif ligand-2), a known autophagy inhibitor and pro-survival factor that is involved in the suppression of starvation-induced macro-autophagy in PCa cells, and thus can enhance the population of tumor-associated macrophages (TAMs) in prostate tumors, in vivo [144]. This suggests the potential clinical significance of bacterial infection or TLR4-activation by LPS in PCa pathogenesis. LPS has been found to play a critical role in the migration, invasion, lymphangiogenesis, and lymph node metastasis of colorectal cancer, thus providing evidence that LPS can increase VEGF-C secretion to promote cell motility and lymphangiogenesis via TLR4- NF-κB/JNK signaling [145]. Similarly, LPS was reported to promote the migration and invasion of colorectal cancer by upregulating VEGFR-3 expression via increased binding of NF-κB to the promoter of VEGFR-3 [145]. In another study, LPS was found to activate inflammasomes in cancer cells through NF-κB signaling and promote metastasis through glycolysis-enhanced activation of the NF-κB/Snail/HK3 signaling pathway. The administration of Metformin was demonstrated to successfully inhibit this pathway [146,147]. LPS acts as a tumor promoter in esophageal squamous cell carcinoma by the epigenetic induction of cancer cell stemness through the activation of an LPS-TET3-HOXB2 signaling axis [148].

The role of LPS-induced inflammation in prostate enlargement or BPH is emerging. LPS-treated prostate epithelial cells were shown to become highly proliferative with increased IL-1β, IL-6, and TNF-α cytokines and elevated HIF-1α levels. The overexpression of HIF-1α by these cells induced the expression of Twist, an initiator of epithelial–mesenchymal transition (EMT) [149]. Since BPH is often accompanied by inflammation, it can be suggested that HIF-1α could drive BPH or prostate enlargement under the influence of LPS-mediated inflammatory signaling. Conversely, natural HIF-1α inhibitors, such as ascorbate and curcumin, have been reported to attenuate EMT and prostate hyperplasia both in vivo and in vitro, therefore providing a promising target for preventing the switch from prostatitis to BPH or PCa.

Unlike LPS, lipoteichoic acid (LTA) is found in the cell wall of gram-positive bacteria organisms. LTA was originally derived from the term “Teichoic acid”, which is used to describe the two polymers found in the wall of gram-positive bacteria: wall teichoic acid (WTA) and lipoteichoic acid (LTA). Originally, teichoic acid was derived from “Techois”, a Greek word for “wall”, after it was discovered in 1958. LTA is an important component of the cell wall in gram-positive bacteria and has been shown to play an essential role in their virulence and homeostasis [150].

Structurally, LTA is an alditol-phosphate polymer linked to the cell membrane via a lipid anchor (Figure 7B). Five types of LTAs can be found in the cell wall of gram-positive bacteria. The Type 1 LTA is made up of a simple unbranched polyglycerol phosphate backbone, while the others (type II–V LTAs) have more complex structures [151,152]. Functionally, LTAs have been linked to inflammation when released in tissues or organs by promoting the release of cytokines from innate immune and epithelial cells through their interaction with TLR2. Both LTA and WTA are targets of the immune system [150]. Innate immune responses via activation of both TLR2/TLR6 and TLR2/TLR1 heterodimers have been described when stimulated with highly purified LTA from *Staphylococcus aureus* and *Streptococcus pneumoniae* [153].

LTAs are responsible for the stimulation of IL-12 by Gram-positive bacteria in human monocytes through the activation of TLR3 and its downstream signaling axis. In a study, PCa cell lines such as DU145, LNCaP, and PC3 were observed to undergo proliferation when stimulated with LTAs [154]. Similarly, at low concentrations, LTAs were also shown to induce a significant increase in cellular proliferation of non-small-cell lung cancer cells (NSCLC) as well as time- and dose-dependent increase in IL-8 secretion [155]. Contrariwise, LTAs from non-pain-inducing (NPI) *Staphylococcus epidermidis* have been experimentally shown to increase the expression of CTLA4-like ligands, PDL1 and PDL2, by prostatic CD11b+ antigen-presenting cells, therefore revealing a potential therapeutic option for treatment of chronic prostatitis-associated pain [156].

### 9.5. Mechanisms of Activation of PRRs by DAMPs

TLRs can be activated by both exogenous and endogenous DAMPs. TLR1 can sense β-defensin-3; TLR2 is activated by heat shock proteins 60 (HSP60), HSP70, HSP90, GP96, HMGB1, HMGB1-nucleosome complex, β-defensin-3, surfactant protein A/D, eosinophil-derived neurotoxins, antiphospholipid antibodies, serum amyloid, bi-glycan, versican, and hyaluronic acid fragments. TLR3 can sense endogenous mRNAs, while TLR4 is activated by HMGB1, fibronectin EDA, tenascin-C, fibrinogen, surfactant protein A/D, β-defensin-3, HSP60, HSP70, HSP22, GP96, S100A8 (MRP8), S100A9 (MRP14), neutrophil elastase, antiphospholipid antibodies, lactoferrin, heparan sulfate fragments, hyaluronic acid fragments, serum amyloid A, oxidized LDL, and saturated fatty acids. TLR7 is activated by antiphospholipid antibodies and ssRNA, while TLR8 is activated by antiphospholipid antibodies and single-stranded RNA (ssRNA). TLR9 is activated by IgG-chromatin complexes [142,157]. TLR1, TLR2, TLR4, TLR5, TLR6, and TLR11 are usually located on the cell membrane, while TLR3, TLR7, TLR8, and TLR9 are found compartmentalized in the endosomes or lysosomes of cells.

As seen in Figure 5, inflammatory signaling via the TLR pathway begins with the activation of the TLRs by DAMPs, followed by the recruitment of MyD88 and TIRAP via the cytoplasmic Toll/IL-1 receptor (TIR) domain. TLR3 and TLR4 can delay the activation of NF-κB by signaling through the MyD88/IRAK1-independent pathway [111]. Following activation of these adaptor proteins, MyD88 recruits IL-1 receptor-associated kinase-4 (IRAK4) to TLRs through the interaction of the death domains of both molecules. IRAK1 is then activated by phosphorylation and autophosphorylation, which results in its binding to the TRAF6 protein. This leads to the formation of a complex made up of TGF-β-activated kinase 1 (TAK1), TAK1-binding protein 1 (TAB1), TAB2, and TAB3. The activation of this complex triggers the activation of both the MAP kinases (JNK1/2, ERK1/2, p38, and MAPK isoforms) and the NF-κB signaling pathway, which is important for cell survival, cell growth, and cytokine production [158]. The recruitment of TOLLIP and IRAK3/M allows for their interaction with IRAK1, which then negatively regulates the TLR-mediated signaling pathways. Another mode of regulation for these pathways includes TRIF-dependent induction of TRAF6 signaling by RIP1 and negative regulation of TIRAP-mediated downstream signaling by ST2L, TRIAD3A, and SOCS1 [159].

All TLRs except TLR3 signal through the MyD88 pathway, while TLR4 can signal through both My88-dependent and MyD88-independent pathways. MyD88-independent pathways can be activated via TRIF and TRAF3, which recruits TBK1 and triggers the phosphorylation of IRF3 and expression of interferon-β [111]. The ligation of PRRs to different TLR ligands can stimulate intracellular signaling cascades that may lead to the expression and secretion of numerous pro-inflammatory mediators, including cytokines and chemokines, as well as the regulation of important downstream cellular signaling pathways [160]. It is worth noting that the nature and degree of activation or regulation of the TLR signaling pathways will depend to a large extent on the specific PRR that has been activated and the type of DAMP. For instance, cancer cells have been demonstrated to release DAMPs while undergoing cell death due to apoptosis or necrosis following exposure to chemotherapy or radiotherapy [117]. This leads to persistent activation of PRRs and chronic inflammation. which promotes cancer proliferation and chemoresistance, and enhances cancer invasion and metastasis by upregulating the expression of pro-inflammatory cytokines, metalloproteinases, and integrins.

### 9.6. Oncogenic Role of TLR Signaling in Prostate Carcinogenesis

Previous studies on TLR signaling were heavily focused on understanding their roles in immune response observed in autoimmune and inflammatory chronic diseases, and to some extent on tumor-associated immune response [161]. While this is very significant, less attention has been dedicated to elucidating the molecular mechanisms regulating intracellular inflammation signaling in tumor cell phenotypes, such as CSCs, NE cells, and differentiated cancer cells, among others. Recent studies show that TLRs are differentially expressed, not just in immune cells, but also in epithelial cells, and that tumor cells including PCa cells can upregulate or downregulate diverse TLRs under different microenvironmental conditions or stimuli [154,162].

A considerable number of studies have shown that gene polymorphism and aberrant activation of some TLRs (e.g., TLR2, TLR3, TLR4, TLR5, TLR7, TLR8, and TLR9) can regulate cell proliferation, survival, apoptosis, and migration/invasion of tumor cells [163]. However, the molecular pathways linking TLR activation to these functions are not fully understood [164]. Upregulation of TLR3, TLR4, and TLR9 in prostate biopsies and histological tissues was found to be predictive and associated with biochemical recurrence in PCa patients [165]. Importantly, multiple cell signaling mechanisms—both canonical and non-canonical—are involved in TLR-mediated inflammatory signaling, and their activation may be dependent on cancer cell type, source of stimuli (either pathogenic or sterile), degree or strength (either acute or chronic inflammation) of stimulation, activated TLR or downstream signaling pathway, and genomic alterations of inflammatory and cell survival genes, among other factors [164,166,167].

TLR signaling has also been described as regulating the activities of some downstream cell survival signaling pathways, such as phosphoinositide-3-kinase (PI3K) and mitogen-activated protein kinases (MAPK), as well as the activation of transcriptional factors, such as nuclear factor kappa-B (NF-κB), interferon response factor (IRF), cAMP response element-binding protein (CREB), and activated protein 1 (AP-1) [168]. Many of the above molecules play important oncogenic roles in diverse tumors, including breast cancer, hepatocellular carcinoma, multiple myeloma, leukemia, and PCa, and also regulate the innate immune effector functions [169,170,171]. Unfortunately, ambiguous outcomes have been reported in terms of activation or repression of TLRs in PCa cells [172].

Activation of PI3K and its downstream kinases (i.e., AKT/PKB and mTOR) are already known to regulate cell proliferation, survival, metabolism, and cytokine production, especially in innate immune cells and tumor cells. However, reports on the effect of PI3K inhibitors on inflammation signaling have been unclear [173]. PI3K has been shown to regulate TLR signaling in both positive and negative ways depending on cell type and other factors, such as the class of PI3K being inhibited or altered. It was shown that PI3K inhibitors including LY294002 along with TLR activation can reduce NF-κB expression/signaling, while another study demonstrated an enhanced pro-inflammatory gene expression in innate immune cells [174]. These effects, though contradictory, may be associated with the feedback loop/mechanism between TLR signaling and PI3K pathway signaling. Another puzzle yet to be solved in prostate carcinogenesis is understanding the specific role of different isoforms of PI3K and AKT in TLR signaling and how their feedback mechanisms contribute to PCa progression.

Some studies on CSCs isolated from other tumors suggest that TLR-mediated inflammation signaling may play an important role in cancer stemness and differentiation [175]. For example, glioma cancer stem cells (GCSCs) were found to evade innate immune suppression of self-renewal through reduced TLR4 expression. However, forced induction or transient overexpression and activation of TLR4 signaling resulted in significant suppression of CSC properties, leading to enhanced GCSC differentiation and growth of non-stem cell phenotypes [176]. The PI3K/AKT pathway is activated in mesenchymal stem cells upon LPS stimulation, which protects them against serum-deprived oxidative stress and apoptosis. PI3K/mTOR inhibitors, such as NVP-BEZ235 and LY294002, were also found to significantly inhibit sphere formation and tumorigenicity in unstimulated PCa cells [177].

Compared to PCa, a significant difference in the expression pattern of TLR3, TLR4, TLR5, TLR7, and TLR9 has been reported for BPH [178,179]. In a study, lower expression of TLR3 has been associated with PCa recurrence and was observed in some PCa tissues compared to BPH samples [180]. Conversely, higher expression of TLR9 has also been described in the epithelium and stroma of PCa tissues compared to BPH tissues [178]. Further research is needed to clarify which of the TLRs play the most significant role in BPH and how we can effectively target them in BPH without altering their physiological function in innate immune signaling.

### 9.7. Oncogenic Significance of Inflammasomes and NOD-like Receptor (NLR) Signaling 

Another notable route by which “danger” signals could trigger prolonged inflammation is through the activation of inflammasomes. An inflammasome is a group of multimeric proteins that form complexes made up of any of the Nod-like receptor (NLR) family members, including NLRP1, NLRP3, NLRC4, NLRP6, NLRP12, and DNA sensor AIM2 (Absent in melanoma 2), Apoptosis-associated speck-like protein containing a carboxyterminal CARD (ASC), and Procaspase-1 [181]. In other words, Inflammasomes are multiprotein complexes formed when one of the NLR protein family (NLRP1-14)/NACHT Leucine-rich-repeat and pyrin domain-containing protein (NALP1-14), Absent in Melanoma 2 (AIM2), and PYRIN, interact with an adaptor, such as the ASC, to initiate downstream activation of inflammatory caspases. For instance, the activation of caspase-1 triggers the cleavage, maturation, and secretion of pro-inflammatory cytokines, such as interleukin-1β (IL-1β) and interleukin-18 (IL-18), which promote inflammation and regulate immune responses [182].

Inflammasomes are usually activated by DAMPs and PAMPs. Examples of DAMPs recognized by inflammasomes include crystalline irritants (e.g., uric acid, silica, asbestos, alum, amyloid β, cholesterol crystals, and monosodium urate), extracellular ATPs, mitochondria ROS (mtROS), mtDNA, hyaluronan, calreticulin, cathepsins, and HMGB1, among others. Some of the PAMPs recognized by inflammasomes include lethal bacterial toxins from *Toxoplasma gondii* and *Bacillus anthracis*, bacterial pore-forming toxins, LPS from gram-negative enterobacteria, bacterial muramyl dipeptides, Rho GTPase modifying toxins, flagellin from *Salmonella typhimurium*, *Pseudomonas aeruginosa*, *Legionella pneumophilia*, and *Shigella flexneri*, and cytosolic dsRNA [183,184]. Bacterial and viral organisms, such as *Staphylococcus aureus*, *Saccharomyces cerevisiae*, *Candida albican*, Influenza viruses, and Adenoviruses, have been reported to induce the activation of the inflammasome directly or indirectly [185]. Inflammasomes can also trigger the secretion of a myriad of leaderless proteins that regulate cell proliferation, apoptosis, cell survival, and tissue repair [185,186,187]. The specific type of inflammasome complex formed per stimuli will be dependent on the NLR family member involved and the interacting adaptors.

Despite being referred to as the master inflammatory regulator, research into the roles of inflammasomes in PCa is still scanty and much remains unknown about their significance in PCa initiation and progression, as well as in other prostate conditions [12]. However, from the few studies already published in the scientific literature, we now know that some of the components of the inflammasome complex, especially NLRP3, AIM2, ASC, caspase-1 (CASP-1), IL-1β, and IL-18, have double-edged effects on inflammation-mediated tumorigenesis [12,188]. For example, CASP-1 can induce pyroptosis, an inflammatory cell death that is accompanied by the secretion of IL-1β and IL-18, therefore triggering a localized inflammation [182]. AIM2 and NLRP3 have been found to induce apoptosis through the activation of caspase-8 (CASP-8) that interacts with the inflammasome complex via ASC. However, single nucleotide polymorphisms (SNPs) and genetic variations in IL-1β have been linked with an increased risk of PCa development and recurrence [66,189].

The pro-inflammatory and tumorigenic effects of cytokines, such as IL-1β and IL-18, have been demonstrated by some studies hence, understanding the mechanistic regulation and upstream signaling of inflammasomes in prostate tumor phenotypes, including PCSCs, NEPCs, and CRPCs, will further help in developing novel therapeutic strategies to efficiently target aberrant inflammasome signaling in PCa [190,191].

### 9.8. Association of Interleukin-1 Receptor (IL-1R) Family Signaling with Chronic Inflammation and Carcinogenesis

Interleukin-1 (IL-1) cytokine receptors have a wide range of pathophysiological functions in autoimmune, cancer, and inflammatory conditions. The history of Interleukin (IL) goes back to the 1940s when the term was first coined and described in the “pathogenesis of fever” by Menkin and Beeson [192]. Follow-up discoveries have led to the use of the term to describe soluble factors secreted by epithelial, innate, and adaptive immune cells. Genetic and biochemical studies have demonstrated that IL-1 receptor (IL-1R)-mediated signaling involves a cascade of kinases organized by multiple adapter molecules such as MyD88 and the IRAKs into signaling complexes, leading to activation of the transcription factor NF-κB [193,194]. Currently, there are about 12 members in the IL-1 cytokine family, including IL-1α, IL-1β, IL-1Rα, IL-18, IL-18BP, IL-33, IL-36α, IL-36β, IL-36γ, IL-36Rα, IL-37, and IL-38). Some of these have proinflammatory functions (IL-1α, IL-1β, IL-18, IL-33, IL-36α, IL-36β, and IL-36γ) while others have anti-inflammatory functions (IL-1Ra, IL-36Ra, IL-37, IL-18BP, and IL-38). They also have similar structural conformation and share co-receptor binding that enables them to exert their biological functions [195,196]. The IL-1 cytokines are made up of three subfamilies: IL-1 subfamily, IL-18 subfamily (IL-18, IL-18BP, IL33), and IL-36 subfamily (IL-36α, IL-36β, IL-36γ, and IL-37) [197].

Many prostate samples express immunoreactions to IL-1β and IL-1R1. In PCa samples of high and low Gleason grades, IL-1α and IL1-R1 have been demonstrated to induce cell proliferation [198]. IL-1β expression is also a common finding in human PIA lesions in the prostate. Exposure of mouse prostate to soluble IL-1β elicited acute and chronic inflammation, including reactive epithelial cells, increased expression of downstream inflammatory proteins and cytokines, altered glandular architecture, and chemoresistance to doxycycline [199]. In a comparative analysis study of RNA sequencing data from AR+ LNCaP PCa cell line versus AR− PC3 PCa cell line, IL-1 signaling genes were found to be overly expressed in LNCaP cells but constitutively expressed in PC3 cells [200], thus showing that IL-1 cytokines signaling through MyD88 and IRAK1/4 adapters contribute to PCa cell survival and tumorigenicity in an inflammatory tumor microenvironment. Similar observations have been found in other tumors; for example, in melanoma cell lines, the IL-1α production was associated with resistance to antiproliferative compounds [201]. In human breast carcinoma, IL-1α, IL-1β, and IL-1Rα were shown to be highly expressed and found to promote tumor progression.

Interestingly, researchers found that patients with a high Gleason grade showed a complete absence of IL-1β expression [198]. These results indicate that the IL-1 family plays a significant role in promoting inflammatory responses in many cancers. This justifies the need for more studies geared towards targeting the downstream adaptors and regulators of the IL-1 signaling pathway, especially the role of IRAKs in prostate diseases. It is important to know whether their aberrant signaling, methylation, or genetic alteration contributes to the transition of chronic prostatitis to BPH or PCa.

### 9.9. Role of IRAKs in Inflammation Signaling and Carcinogenesis

IL-1 receptor-associated kinases (IRAKs) are a unique family of serine-threonine kinases that play a critical role in regulating the inflammatory signaling of two upstream receptor families, the IL-1 receptor family (IL-1R, IL-18R, IL-33R) and Toll-like receptor family (TLR1–10). IRAKs are located at the intercept between the upstream (TLRs and IL-1Rs) and the downstream (TRAF6 and NF-κB transcription factors) inflammatory signaling proteins [202]. The IRAK gene family is made up of four genes (IRAK1, 2, 3/M, and 4) that are expressed canonically and sometimes non-canonically in both epithelial and immune cells, except for IRAK3, which is overly expressed in monocytes or macrophages compared to others [203]. IRAKs, among others, have death domains for interaction with one another and neighboring proteins and the kinase domain for catalytic activities [204]. Both IRAK1 and IRAK4 are known to show true kinase activity and have been exclusively explored as targets in a variety of auto-immune and inflammatory diseases: rheumatoid arthritis, diabetes, and systemic lupus erythematosus, as well as in a few cancer cases in the scientific literature [205]. IRAK2 and IRAK3 lack kinase activity due to the absence of aspartate residue in the active site of their kinase domain, also known as the pseudo-kinase domain [206].

Though IRAKs were initially thought to be associated only with the signal transduction activity of interleukin-1 (IL-1), recent studies have demonstrated the significance of this family in the signaling of other members of the Toll/IL-1 receptor family (Figure 8). Unlike IRAK1, 2, and 4, the IRAK3 is a negative regulator of the IL1-R/TLRs-mediated NF-κB signaling pathway, by inhibiting the dissociation of IRAK1 and IRAK4 from the TLR signaling complex after IRAK1 activation, as well as inhibiting the binding of IRAK1 to TRAF6 [204]. The inhibition of IRAK1 by gene silencing and small molecule inhibitors was found to induce apoptosis and a decrease in cell viability in melanoma, gastric cancer, lymphoblastic leukemia, pancreatic cancer, Kaposi carcinoma, cervical cancer, and hepatocellular carcinoma [207,208,209,210,211]. Enhanced angiogenesis and metastasis were observed following the upregulation of IRAK1 in breast cancer [209]. In liver cancer, overexpression of IRAK1 was found to be critical for the maintenance of aggressive tumor-initiating/stem cells [212].

However, in contrast to the above findings, the downregulation of IRAK1 was reported to aid the aggressiveness of papillary thyroid carcinoma [207]. The overexpression of IRAK3 was found to support the aggressiveness of colorectal cancer, while an opposite finding was reported in a study done on melanoma cells. Interestingly, three epidemiological studies have reported a possible link between single-nucleotide polymorphisms (SNPs) in the IRAK4 gene and an increased risk of PCa in Korean, Swedish, and African American men [68,69,213]. The role of structural variation, dysregulation, or methylation of IRAKs and how they contribute to PCa aggressiveness and progression have not been exclusively defined. Moreover, the molecular mechanisms by which dysregulation of IRAK signaling drives therapy resistance, metastasis, and tumor heterogeneity in PCa patients are unknown and yet to be studied.

### 9.10. Molecular Mechanisms of IRAK1 Signaling Transduction

In humans, the IRAK1 protein is encoded by the IRAK1 gene. It is located on chromosome X (Xq28) with a molecular weight of 76,537 Da. IRAK1 is one of the most significant and widely studied members of the IRAK family due to its degree of interaction and importance in regulating several downstream signaling molecules, such as the NF-κB-dependent transcription factors [214]. IRAK1 interacts with virtually all members of the family. Studies have shown that IRAK1 can directly interact with itself as well as with MYD88, IRAK2, IRAK3, and IRAK4. The interaction of IRAK1 with IRAK2, as well as IRAK3, forms a heterocomplex, while its self-interaction results in autophosphorylation [194]. IRAK1, together with IRAK4, are the only two true serine/threonine kinases in this family and they both undergo catalytic activities or kinase signaling upon activation and are associated with the IL1-R and TLR signaling pathways. IRAK1 could also interact and phosphorylate PELI2. Since phosphorylation is well-known as a promoter of ubiquitination, the activation of IRAK4 by MyD88 initiates IRAK4 phosphorylation and autophosphorylation, which subsequently results in the phosphorylation of IRAK1 through pellino-mediated polyubiquitination after prior phosphorylation of Pellino E3 ubiquitin-protein ligases, such as PELI1, PELI1, and PELI3 by IRAK4 [215]. Mal is another upstream molecule that has recently been shown to interact with IRAK1 and IRAK4 via its TIR domain. The overexpression of Mal was found to interact with active IRAK4 and IRAK1, which enhances its degradation [216]. On the other end of the signaling cascade, the polyubiquitinated IRAK1 binds to the ubiquitin-binding domain of IKBKG/NEMO, which forms a bridge between two complexes, IRAK1-MAP3K7/TAK1/TAK2-TRAF6 and the NEMO-IKKA-IKKB complexes, which are released into the cytoplasm. The MAP3K7/TAK1 axis of these bipartite complexes is responsible for the activation of the IKKs, such as the CHUK/IKKA and the IKBKB/IKKB, as well as NF-κB activation and nuclear translocation [166,217].

IRAK4 also plays a part in IRAK1’s downstream signaling activities. For instance, IRAK4 interacts with the death domain of IRAK1 to activate a series of sequential phosphorylation and autophosphorylation of IRAK1 starting at the Thr209 phosphorylation site, which creates a conformational change in the kinase domain. This conformational change creates the impetus for more autophosphorylation to be triggered. The second phosphorylation occurs at the Thr387 phosphorylation site, which is located within the activation loop of the domain and leads to full enzymatic activity [194]. Subsequent multiple autophosphorylations of IRAK1 have been reported in the proline-, serine-, and threonine-rich ProST domain, located between the N-terminal death domain and the kinase domain. The substitution of threonine with alanine at Thr209 has been shown to lead to the formation of a kinase-inactive IRAK1 [218].

Preferentially, IRAK1 could also phosphorylate TIRAP to enhance its ubiquitination and degradation. Interferon regulatory factor 7 (IRF7) could also be phosphorylated by IRAK1 to induce its translocation into the nucleus in order to activate the type 1 IFN transcriptional genes [219]. Upon sumoylation, IRAK1 could also be translocated into the nucleus to phosphorylate STAT3 and induce interleukin-10 (IL-10) expression. Interestingly, the splenocytes from IRAK1-deficient mice failed to exhibit LPS-induced STAT3 serine phosphorylation and IL-10 expression, despite being exposed to LPS [220]. Furthermore, IRAK1 has been demonstrated to undergo degradation within 1 h of its activation, after which time IRAK2 functions to sustain TLR responses [221]. However, another study countered such an assumption with a seminal finding that the ‘disappearance’ of IRAK1 is not caused by its degradation, but by its conversion into slowly migrating phosphorylated and ubiquitylated forms, which are poorly recognized by common IRAK1 antibodies [222]. Thus, it was opined that the unmodified forms of IRAK1 could be fully recovered by treatment with phosphatase or deubiquitinase.

Structurally, IRAK family members have a uniquely similar and conserved domain architecture, divided into four domains, except for IRAK4 (Figure 9). The most proximal is the N-terminal death domain (DD), which is followed by the proline/serine/threonine rich domain (ProST), a C-terminal kinase domain (KD) (in IRAK1 and IRAK4), or pseudo-kinase domain (in IRAK2 and IRAK3), and a C-terminal domain required for the recruitment and activation of the downstream effector TNF receptor-associated factor 6 (TRAF6). IRAK4 does not have the C-terminal domain, but uses the DD for its interaction with IRAK1 to induce its phosphorylation through IRAK1’s DD. All members use the DD for binding or interacting with upstream repressors or activators.

### 9.11. Clinical Significance of Aberrant IRAK1 Signaling in Cancer Progression

Several studies have linked chronic inflammation signaling through the TLR-IRAK pathways to the induction of genetic alterations/instability, tumor initiation, tumor promotion, and tumor progression, including tumor invasion, metastasis, and therapy resistance [223,224,225]. IRAKs, for a long time now, have been thought to have functions independent of the TLR/IL-1 signaling transduction. Indeed, the catalytic activity of IRAK1 is believed to play diverse roles in cell functions other than its innate immunity and inflammation functions [218]. For instance, IRAK1 is highly expressed in multiple cancer cells and is a critical driver of intrinsic tumor resistance to radiotherapy (RT) and chemotherapy [124,226]. Hence, IRAK1 has become an essential protein for cell survival in many human cancer cells. The catalytic activity of IRAK1 has been reported to be important for cancer cell survival following exposure to radiotherapy or chemotherapy. Apart from signaling through the NF-κB pathway, IRAK1 has been demonstrated to promote cell survival through its interaction with the PIDDosome complex (PIDD-RAIDD-Caspase 2) to prevent apoptosis [16].

Essentially, IRAK1 is implicated in the pathogenesis and tumor progression of multiple cancers, including breast cancer, hepatocellular carcinoma, T-acute lymphoblastic leukemia, activated B cell-like diffuse large B cell lymphoma, melanoma, Kaposi sarcoma, lung cancer, head and neck squamous carcinoma, endometrial carcinoma, and pancreatic cancer, among others [209,211,224,227,228]. For instance, IRAK1 upregulation causes an increase in tumor growth and metastasis of breast cancer. In triple-negative breast cancer (TNBC), IRAK1 overexpression was found to positively correlate with poor prognosis and the mechanism of TNBC growth was suggested to be IRAK1/NF-κB-dependent [209]. High expression of IRAK1 was also observed in many endometrial cancer patients and found to be positively correlated with poor prognosis, higher tumor metastasis, myometrial invasion, and tumor-associated deaths. In hepatocellular carcinoma (HCC), IRAK1 upregulation is associated with poor therapy response, metastasis, and an increase in tumor size. IRAK1 overexpression was found to be a very good diagnostic and prognostic marker in HCC [210]. In myeloproliferative neoplasms (MPNs), IRAK1 was implicated in the overexpression of protumor and pro-inflammatory cytokines, such as IL-6 and IL-8, through aberrant NF-κB signaling.

Furthermore, IRAK1 was overexpressed in 20–30% of patients with myelodysplastic syndromes (MDS) [229]. MDS patients with high expression of IRAK1 show poor survival and tumor progression. The mechanism of IRAK1’s promotion of MDS was due to IRAK1’s ability to activate the NLRP3 inflammasome, which has been implicated in the pathogenesis of MDS. IRAK1 has also been listed as one of the NF-κB-associated genes upregulated in follicular lymphoma and has been shown to be positively correlated with the aggressiveness of the disease. In activated B cell-like diffuse large B cell lymphoma (ABC-DLBCL), IRAK1 overexpression contributes to DLBCL aggressiveness in patients with MyD88 gain-of-function mutation [230]. Similarly, to the other findings, patient-derived T-acute lymphoblastic leukemia (T-ALL) cells also show elevated levels of IRAK1 mRNA [231]. IRAK1/4 inhibitor I, IRAK1 shRNA, and IRAK4 shRNA show antiproliferative effects on T-ALL cells through a mechanism involving destabilization of the anti-apoptotic protein-induced myeloid leukemia cell differentiation protein 1 (MCL-1) [224]. About 42% of melanoma cell lines constitutively express phospho-IRAK1, which has been implicated in the aggressiveness of the disease. IRAK1 is also overexpressed in about 14% of head and neck squamous carcinoma (HNSCC) patients [227]. IRAK1 has also been proposed to play an oncogenic role in Kaposi sarcoma [128].

In conclusion, the high expression and the oncogenic role of IRAK1 in many of these cancers further justifies the need to investigate its role in PCa, as well as the therapeutic potential of targeting IRAK1 in PCa cells. An understanding of the crosstalk between IRAK1 signaling and many other genes or pathways that have been associated with PCa progression and aggressiveness would be a necessary step toward achieving this goal (Figure 10).

### 9.12. Targeting Aberrant IRAK1 Signaling in Cancers

The inhibition of IRAK1 kinase activity has long been shown to modify IL-1 and TLR signaling. The extent of the modification has been suggested to be dependent on the degree of phosphorylation of the kinase and ProST domains of IRAK1, as well as the cell type. The pharmacological inhibition of IRAK1 by IRAK1/4 inhibitor I was shown to have both antiproliferative and antimetastatic effects on hepatocellular carcinoma (HCC) cells [194]. The downregulation of IRAK1 expression in HCC using IRAK1 siRNA was also observed to inhibit tumor growth and enhance sensitivity to cisplatin-induced apoptosis [212]. Pacritinib, another IRAK1 inhibitor, was found to prevent fibrosis formation and to decrease serum levels of cytokeratin 18. This indicates that the suppression of IRAK1 by Pacritinib may be beneficial in preventing premalignant fibrosis [211]. In MDS, the inhibition of IRAK1 by the IRAK1 inhibitor I was shown to block NF-κB activation, thereby interfering with MDS progenitor cell function and growth in an MDS xenograft model [229]. In ABC-DLBCL, the combined administration of ibrutinib with IRAK1/4 inhibitor I provided a synergistic cell-killing effect in MyD88 L265P cell lines and showed enhanced cell killing relative to either agent alone in primary MyD88 L265P cells [230]. In T-ALL, IRAK1/4 inhibitor I, and IRAK1 shRNA show antiproliferative effects on T-ALL cells by destabilizing the MCL-1. IRAK1/4 inhibitor, in combination with vinblastine, improved tumor growth inhibition in a xenograft mouse model and was also found to enhance the apoptosis of melanoma cell lines [231].

Of significance is the relationship between IRAK1 and IRAK4 in many of these studies. A crystallization study on IRAKs shows that the overall sequence identity among the IRAK family members is about 30%, though a 90% similarity was found in the residues lining the ATP-binding pocket between IRAK1 and IRAK4 [202,231]. This may be responsible for the effectiveness of using small molecule inhibitors that simultaneously target IRAK1 and IRAK4 [232]. However, a highly selective inhibitor for IRAK1 will be necessary to better differentiate the individual effect of each protein. Targeting IRAK1 signaling in PCa and other prostate pathologies comes with multiple benefits in impeding chronic inflammatory effects and also dampening the tumorigenic activities of the NF-κB transcriptional factors downstream of IRAK1 in the inflammatory cascade. Below is a diagrammatic sketch of a potential therapeutic strategy for targeting IRAKs and managing advanced inflammation-associated PCa (Figure 11).

### 9.13. Molecular Mechanisms of NF-κB Signaling in Cancer

The NF-κB proteins are a well-known and widely studied family of five transcription factors with great importance in inflammation, immunity, cell survival, growth, development, migration, response to stress, and proliferation [233]. Due to their pleiotropic properties, NF-κB transcription factors regulate, directly or indirectly, the signaling transduction and expression of several hundreds of genes that play critical roles in the biological and molecular functions of cells [234]. This includes cell growth, apoptosis, development, migration, cell cycle, and differentiation, among others. within the body [200]. Structurally, NF-κB proteins utilize their Rel-homology (RHD) domain for dimerization and DNA binding. They can also bind to one another to form homodimers and heterodimers. NF-κB proteins are categorically divided into two groups, the Rel members and the NF-κB precursor proteins [235]. The Rel members are three in number and are known to have a C-terminal transcription activation domain (also known as TAD). Rel members include RelA (p65), RelB, and cRel and are important for the positive regulation of gene expression and signal transduction [236].

The other two NF-κB family subunits are the p105 and p100 precursor proteins, which lack the TAD but have C-terminal ankyrin repeats that prevent DNA binding until they have been processed into their smaller units, p50, and p52, respectively. The lack of TAD by p100 and p105 means they are incapable of activating gene expression or transcription on their own, except when homodimerized with one of the Rel members [237]. RelB is the only member that cannot form homodimers; however, it can bind to form heterodimers. p50 has a strong affinity for RelA, while p52 has a strong affinity for RelB. The p105, a larger subunit of NF-κB1 gene product, is constitutively broken down into a p50, a smaller subunit, through the help of the proteasome. p50 is automatically rendered inactive as a heterodimer with RelA or c-Rel due to its interaction with the inhibitory IκB proteins [238].

Situated upstream to the NF-κB transcriptional factors are the IκB proteins, which function to regulate the activity of the NF-κB (Figure 12). They also possess the C-terminal ankyrin repeats that favor DNA binding inhibition. IκB is regulated by the phosphorylation of the IκB kinases [128]. IκBα, IκBβ, and IκBɛ have been demonstrated to bind to the DNA binding domains of NF-κB proteins. They can facilitate the export of NF-κB proteins out of the nucleus through multiple nuclear export signals (NES), whereas IκBζ and Bcl-3 are specifically located in the nucleus and have a strong binding affinity for only p50 and p52 homodimers [235]. NF-κB proteins are activated through two pathways: the canonical and non-canonical pathways. The canonical pathway is usually closely related to the p50 subunit of p105 and activated by upstream pro-inflammatory receptors, including IL-1Rs, TLRs, and the TNF superfamily, as well as exposure to genotoxic agents. The signal transduction of these receptors recruits and leads to the activation of the IKK complex formed by IKKa, IKKβ, and IKKγ (NEMO). The first two act as the catalytic kinases, while the NEMO acts as the regulatory subunit [239].

On the other hand, the non-canonical NF-κB pathway can be activated by subsets of the TNF receptor family members, such as CD40, LTβR, CD30, RANK, CD27, and BAFF-R, among others. This pathway relies on the phosphorylation-induced processing of p100 to form the smaller subunit, p52 [235]. Unlike the canonical NF-κB pathway, the non-canonical preferentially depends on the upstream activation of NF-κB-inducing kinase (NIK) and IKKɑ through the recruitment of TRAF2 and TRAF3, instead of the trimeric IKK complex [240]. The stabilization of NIK and IKKɑ is important for the downstream activation of molecules involved in this pathway, whereas the degradation of NIK or IKKɑ will hinder the activation of this pathway. NF-κB p65 protein, when in the cytoplasm, is usually inactivated by IκB inhibitory proteins [241]. However, upon activation, IκBs become phosphorylated and degraded via ubiquitination and proteasomal degradation, which favors the translocation of p65 to the nucleus, where it regulates the transcription of diverse genes involved in cell survival, invasion, and metastasis [242]. Several key genes are regulated by NF-κB p65, including VEGF, MMP-9, BCL-2, IL-6, cyclin D1, IL1β, IL-8, IL-18, and BAD, among many others [243].

### 9.14. Role of NF-κB p65 (RELA) Signaling in Prostate Carcinogenesis

Several members of the NF-κB family play major roles in cancer initiation and progression. Changes in the expression levels of nuclear factor κB subunit 1 (NF-κB1), nuclear factor κB subunit 2 (NF-κB2), REL proto-oncogene nuclear factor κB p65 subunit (RELA), and RELB proto-oncogene nuclear factor κB subunit (RELB) and c-REL proto-oncogene nuclear factor κB (c-REL) have been associated with tumorigenesis [238]. Constitutive activation of NF-κB family members and their transcription factors has been reported in several tumors, including pancreatic cancer, breast cancer, ovarian cancer, and PCa [209,244,245]. The overexpression of several NF-κB transcription factors in the nucleus of PCa cells has been associated with chemoresistance and clonal evolution of aggressive cancer phenotypes, as well as cancer growth, survival, stemness, angiogenesis, and metastasis [171]. Studies have also demonstrated that aberrant activation of NF-κB transcription factors promotes the development of castrate-resistant PCa subtypes [246].

Specifically, the overexpression of NF-κB p65 has been proposed as a prognostic biomarker for the identification of patients with aggressive PCa [200]. Similar to the other NF-κB subunits, the trans-localization of NF-κB p65 into the nucleus of PCa cells positively correlates with most PCa clinical endpoints, such as metastases, biochemical recurrence, progression, and deaths [246]. Thus, inhibition of NF-κB p65 has the potential to become beneficial for treating CRPCs. Since IRAK1 is also important for NF-κB p65 activation, we propose that inhibition of IRAK1 may be another therapeutic strategy to regulate NF-κB p65 activation, as well as to simultaneously impede pro-tumorigenic and pro-inflammatory signaling in PCa.

## 10. Prostate Tumor Heterogeneity and Progression

Several hallmarks have been identified during PCa progression (Figure 13). Although early-stage PCa can be managed with radical prostatectomy, radiotherapy, chemotherapy, and ADT, almost half of PCa survivors will eventually develop a biochemical recurrence in the form of CRPC [247]. There is currently no curative treatment for men with metastatic PRAD and reoccurring CRPC [248]. Therefore, complications due to the recurrence of metastatic CRPC are suggested as the major cause of death among PCa patients/survivors. The insurgence of CRPCs has also been linked to various androgen-resistant phenotypes in the tumor microenvironment. For instance, PCa stem cells (PCSCs) are resistant to ADT and radiotherapy and are known to be highly immunosuppressive [249]. Cumulatively, aggressive behaviors include PCa stemness, neuroendocrine progression, and recurrence [248,250,251].

### 10.1. Role of Cancer Stemness in Prostate Cancer Progression

PCSCs, like other CSCs, are known to be highly tumorigenic, metastatic, and chemo-radioresistant, undergo self-renewal/pluripotency, metabolic reprogramming and, most importantly, evade immune surveillance and attack. Cell survival pathways, such as the TLR/IL1-R/NF-κB, PTEN/PI3K/AKT/mTOR, and RAS/MAPK pathways, are important players in the maintenance of PCSCs and their progenitors in castration-resistant prostate tumors [252,253,254]. The co-activation of PI3K/AKT/mTOR and RAS/MAPK signaling in PCSCs favors tumorigenicity and invasiveness. Deletion mutation of PTEN, as well as amplification of the androgen receptor (AR), can drive PCa progression through the emergence of several CRPC phenotypes [255]. Many small molecule compounds that act as pathway inhibitors have been tested in pre-clinical studies, but only a few made it through clinical trials due to unsolicited toxicity and off-target/side effects [256]. There is no doubt that developing therapeutic strategies to target, effectively and successfully, the cell survival and cancer stemness signaling pathways in PCa will be beneficial to the early elimination of PCSCs from the tumor microenvironment.

Stemness is a term used to describe the phenotypic nature and functional characteristics of pluripotent, stem-like, and progenitor cells capable of undergoing self-renewal, proliferation, and trans-differentiation through a process known as plasticity [257]. Importantly, stemness is an indispensable feature of CSCs and has been confirmed in vitro and in vivo. Several cell-based markers have been adopted to identify, isolate, and differentiate PCSCs from other tumor subpopulations [258]. PCSCs can be isolated from prostate tumor samples and cell lines through fluorescent-activated cell sorting (FACS) and magnetic-activated cell sorting (MACS) techniques [259]. The commonly used CSC markers are CD44, CD133, α_2_β_1_ integrin, and CD24, among others [260]. The combination of two or more of these markers is more accurate and efficient in identifying and isolating PCSC populations and progenitors. For example, a study showed that CD133^+^ PCSCs isolated from PCa tissues also show higher expression of β1 integrin and α_2_β_1_ integrin [256]. In another study, PCSCs were putatively identified in primary prostate tumors using a combination of cell markers: CD44^+^/α_2_β_1_^high^/CD133^+^ [261]. Interestingly, stem-like cells expressing CD44^+^/α_2_β_1_^high^/CD133^+^ were found to display increased tumorigenesis and self-renewal relative to CD44^−^/α_2_β_1_^low^/CD133^−^ cells. Surprisingly, non-adherent sphere-forming PCSCs with CD44^+^/α_2_β_1_^high^/CD24^+^ phenotypes were also isolated from both androgen-sensitive (AR^+^) and CRPC (AR^−^) cell lines grown in serum-free media and growth factor-reduced Matrigel matrix. They were shown to display enhanced tumorigenicity in vivo [252].

Apart from the above-listed CSC markers, PCSCs have also been shown to display normal stem cell transcription factors or markers, such as NANOG, SOX2, KLF4, OCT4/POU5F1, c-MYC, ALDH1, CXCR4, TRA-1-60, CD151, CD166, SCA1, LIN28B, and BMI1, among others [256]. In addition to their tumorigenicity and high metastatic rate, PCSCs isolated from either AR^+^ or AR^−^ PCa cell lines were observed to undergo epithelial-to-mesenchymal (EMT) phenotypic transition, by displaying mesenchymal surface markers instead of epithelial markers. Many other CSC signaling pathways, including Wnt/β-Catenin, Notch, TGF-β, and Hedgehog, have been studied, shown to play important roles in the maintenance of PCSCs, and may contribute to PCa progression and recurrence [252,262].

### 10.2. Linking Chronic Inflammation to Prostate Cancer Stemness

As mentioned above, CSCs have similar characteristics to normal stem cells (SCs). CSCs are known as a small subpopulation of undifferentiated cells within a tumor that can undergo self-renewal and differentiation and are highly tumorigenic [260]. A distinctive characteristic that is restricted to CSCs is tumorigenicity, which means that these cells can form new tumors. As we know, chronic inflammation comes with an increased risk of cancer development, but chronic inflammation could also affect cancer stemness and normal stem cell functions. There have been some studies aimed at understanding the contribution of prolonged inflammatory signaling to the formation of CSCs [254]. NF-κB-mediated proinflammatory signaling has been linked with tumor initiation, tumor progression, and metastasis, as well as regulation of genes associated with cancer cell proliferation, apoptosis, survival, and invasion [263]. Increased NF-κB activity and inflammatory responses in CSCs have been associated with aberrant TLR signaling during the CSC enrichment in tumors [254]. NF-κB-mediated inflammatory responses can also support CSC expansion. For instance, increased NF-κB activation and nuclei translocation of p65/RelA has been reported in glioblastoma, leukemia, and ovarian CSCs compared to the non-CSCs [264,265,266]. Other studies have also demonstrated the ability of NF-κB activation to induce the expression of stemness-associated genes and EMT regulators such as SNAIL, SLUG, and TWIST1 in cancer cells, thereby generating a CSC phenotype [254]. CD44^+^/CD24^−^ breast cancer cells have also been shown to upregulate NF-κB compared to CD44^−^/CD24^+^ breast cancer cells [267].

Many researchers believe that CSCs are responsible for the heterogeneous nature of tumors and that they contribute to the cancer progression or biochemical recurrence in patients or survivors after being in remission for several months or years [268]. In the prostate, SCs have been isolated from both the basal and luminal cell layers, and hence, could undergo oncogenic transformation [269]. TRA-1-60^+^/CD151^+^/CD166^+^ PCSCs in human prostate tumors have been demonstrated to display increased NF-κB activity and proinflammatory gene expression [270]. An NF-κB inhibitor called parthenolide has been shown to significantly induce apoptosis in both CD133^+^ CSC and normal progenitor cells [271], thus implying that the transformation of SCs to CSCs in the prostate may be a critical process in the initiation of PCa, and this transformation could be impacted or enhanced by the pro-tumorigenic effects of chronic inflammatory signals [272]. It will be interesting to explore and elucidate the molecular mechanisms and roles of other inflammatory molecules in PCa stemness and if these molecules are therapeutically targetable to prevent PCa progression and recurrence/relapse.

### 10.3. Role of Neuroendocrine Differentiation in Prostate Cancer Progression

Neuroendocrine (NE) cells are very similar to nerve cells in many ways. NE cells function by receiving signals from the nervous system and, in response, they create and release hormones that assist in many of the body’s functions [273,274]. Some of the hormones released by NE cells that assist in the body’s functions include insulin, serotonin, epinephrine, growth hormones from the pituitary gland, and others [275]. Secretions from NE cells assist in diverse body functions, such as those involved in controlling the amount of air and blood flowing through the lungs, blood pressure and heart rate, the amount of glucose within the blood, and the growth of muscles and bones, along with their development [169]. When NE cells are found in tumors, they also secrete factors that act as a stimulant to the survival, growth, and metastatic potential of cancer cells [276]. NE cells are naturally occurring in the prostate gland and are believed to support growth and differentiation. An important characteristic to keep in mind about tumorigenic NE cells is that they lack the expression of androgen receptors (AR) and are resistant to hormonal therapy [277].

NE prostate tumor is a rare and very aggressive tumor that originates from mutated or genetically altered NE cells of the prostate [278]. Studies have shown that NE PCa cells (NEPCs) or NE-like PCa cells can be formed from luminal and basal PCa cells that have been subjected to prolonged androgen deprivation or stressful conditions through a process known as Trans-differentiation [279]. This concept has been proven by the formation of NE-like cells—with prominent neurites and NE secretions—following prolonged maintenance of androgen-dependent LNCaP cell line in a low serum medium culture (supplemented with <5% fetal bovine serum) for at least a month and has been confirmed in our current study [75].

Clinically, NEPC is considered an aggressive or lethal variant of metastatic CRPC (mCRPC), enriched in PCa patients following ADT [278]. NEPC is subdivided into the following subtypes: (i) Small cell prostate carcinoma (SCPC), due to their resemblance with the small cell carcinoma of the lung, which consists of poorly differentiated small tumor cells with minimal cytoplasm, nuclear molding, fine chromatin pattern, extensive tumor necrosis/apoptosis, and a brisk mitotic rate, highly metastatic, highly proliferative, and chemo-resistant to most therapies; (ii) NE carcinoma of the prostate (NEC-P); (iii) Large cell NE carcinoma; (iv) Traditional PRAD with neuroendocrine differentiation (NED); (v) PRAD with Paneth cell NED; (vi) Mixed (small and/or large cell) NE carcinoma-acinar adenocarcinoma; and (vii) Low-grade carcinoid of the prostate. Due to the aggressiveness and lethality of this disease, the overall survival is typically less than 1 year from the time of diagnosis [279,280].

Notably, NEPCs share certain molecular characteristics with PCSCs, such as lack of androgen receptor (AR−), retinoblastoma gene (RB−), and prostate-specific antigen (PSA^−^), while they differ in specific expression of neuronal differentiation markers, such as chromogranin A, neuron-specific enolase (NSE), synaptophysin, tubulin beta III (TUBB3), TTF-1, and CD56 (NCAM) by NEPCs [262,281]. When comparing the NE cells in PCa tissue and those found in benign tissue, it is visible that the NE cells in the PCa tissue are morphologically different and display both epithelial markers and NE markers [278].

Unlike PCSCs and CRPCs, NEPCs produce neurosecretory granules that are rich in various neuropeptides and hormones (e.g., bombesin, parathyroid hormone-related peptides, serotonin, calcitonin, and adrenomedullin, among others), biogenic amines, and growth factors (e.g., VEGF) implicated in the regulation of cellular homeostasis, proliferation, differentiation, and tumorigenesis [282]. Other commonly observed genomic alterations in NEPCs include the copy number amplification and overexpression of genes, such as Aurora kinase A (AURKA), Neuroblastoma-derived MYC (MYCN), Enhancer of Zeste homolog 2 (EZH2), and Cyclin D1 (CCND1), UBE2C, NTS, MDK, NPPB, NTSR1, RB1, FOXA2, REST, and SRRM4 relative to prostate adenocarcinoma [283,284,285,286]. N-MYC is closely related to the AURKA and essential for NEPC maintenance in prostate tumors. Interestingly, the destabilization of N-MYC by AURKA inhibitors has been reported to induce tumor cell deaths [287,288].

NED occurs when the prostate tissues contain multiple cells that display either endocrine, neuronal, or both features [289]. NEPCs can either be the main component of the tumor or a subpopulation of ADPCs/PRAD or CRPC dominant tumors. Morphologically, NEPCs appear as small cells with pronounced dendritic processes, like those found in neuronal cells. NEPCs specifically have been demonstrated to share some molecular similarities with PCSCs by expressing stem/progenitor cell markers, such as CD44, p63, and c-Kit on their surface, which can be used to distinguish them from the other NE tumors found in the body. NE markers are commonly linked to CRPC, usually when cancer becomes metastasized. A study by Lee and colleagues examined premalignant prostatic intraepithelial neoplasia and showed that AKT1 is associated with the expression of MYCN in normal prostate basal cells which contributes to PCa development and progression. AKT1/MYCN-mediated progression was also associated with the development of invasive metastatic castration-resistant prostate tumors [290].

In the past, NE tumors within the prostate have been treated and managed through cisplatin-based chemotherapy regimens, although additional research and understanding of the gene markers and their role in the progression of cancer is needed to produce novel targeted therapies [288]. Unfortunately, about 50% of metastatic CRPC patients are expected to eventually develop NEPCs following ADT, including about half with pure small-cell histology and half with a hybrid phenotype between adenocarcinoma and small-cell carcinoma [284]. Therefore, understanding the underlying mechanisms, through which prostate tumors gain aggressive PCa phenotypes is critical for the development of novel therapeutics.

### 10.4. Linking Chronic Inflammation to Neuroendocrine Differentiation

There have been multiple attempts to link chronic inflammation to NED. It has been reported that numerous inflammatory cytokines play a key role in inducing NED within PCa cells and, as a result of this NED, causes the release of numerous cytokines, which stimulates the progression of PCa [276]. NED in prostatic adenocarcinomas is now regarded as one of the early markers of recurrence or progression to castration-resistant phenotypes, characterized by poor prognosis and limited treatment options. Though the maintenance and enrichment of NEPCs and PCSCs are associated with tumor progression, castration resistance, and tumor recurrence, the connection between chronic inflammation signaling and these hallmarks is still not fully understood [25,278]. In a study to characterize the molecular differences between metastatic biopsies from patients that have been histologically characterized as either CRPC (adenocarcinoma) or NEPC, marked epigenetic differences were found between CRPC and NEPC using genome-wide DNA methylation meta-analysis, while significant genomic overlap between both CRPC subtypes was observed using whole-exome sequencing [291]. Thus, this finding suggests that NEPCs can clonally evolve from CRPC (adenocarcinoma) and that an epigenetic modifier such as chronic inflammation may promote the maintenance and development of castration-resistant phenotypes. In two other studies, androgen-dependent LNCaP cells were found to acquire NE characteristics in adaptation to conditional and microenvironmental changes in culture in vitro, after long-term androgen deprivation, increased intracellular cyclic AMP (cAMP) levels, and stimulation with interleukin-1β (IL-1β) and interleukin-6 (IL-6) or a combination of two or more of the above [275,292,293].

Perhaps, one of the most exciting justifications for our study is the discovery that exogenous IL-6, when administered alone or in combination with cyclic AMP-dependent protein kinase (PKA) signaling agonists/activators, such as epinephrine and forskolin, can promote the acquisition of NE phenotype and increase the production of mitogenic neuropeptides by transformed LNCaP cells, dose-dependently [292], thus justifying the potential contribution of inflammation signaling—directly or indirectly—to NED or castration resistance. Since their study only involved the administration of exogenous IL-6, we currently do not know whether the IL-6-induced NE trans-differentiation and LNCaP acquisition of castration resistance can also be endogenously triggered by intracellularly derived IL-6 within the cancer cells, or extracellularly from within the tumor microenvironment from other IL-6 producing tumor-associated cells, including macrophages [294]. We believe a co-culture mechanistic study will be a good approach to solving this puzzle.

Interleukin 6 (IL-6) is one of the most extensively studied inflammatory cytokines. Serum levels of IL-6 are usually elevated in mCRPC patients and have been shown to play an important role in driving aggressive PCa phenotypes, cancer progression, metastasis, epithelial-mesenchymal transition (EMT), drug resistance, and cancer cell survival by networking with multiple signaling pathways. Interleukin-6 (IL-6) can also be produced by many tumor-associated stromal cells, including tumor-associated innate immune [295]. Other studies have shown that IL-6 may have both proliferative and inhibitory effects on PCa cells [292]. However, we believe these dual effects may be dependent on many factors, including variations in the doses of IL-6 administered, exposure time, laboratory conditions of growth, and intrinsic genomic predisposition. To understand the underlying molecular mechanism of inflammation-associated cytokines such as IL-6, a better understanding of the roles of their upstream regulators, such as TLR and IRAK signaling in regulating their expression and function in PCSCs, NEPCs, and CRPCs, is important.

### 10.5. Linking Chronic Inflammation to Castration Resistance and Prostate Cancer Progression

ADT is a form of hormone therapy for PCa to slow the growth, or in some cases shrink, the tumor by inhibiting the availability of androgens (male sex hormones) to prostate cells. Prostate tumors are known to be dependent on AR signaling and, when patients are treated with ADT, they often respond initially but soon experience biochemical recurrence of an androgen-insensitive form of PCa, which is known as CRPC [296]. When a patient has CRPC, it means the PCa cells have undergone differentiation and advanced so much that they no longer require androgens to grow. The disease becomes incurable since CRPCs are unresponsive to ADT and other conventional treatment options.

CRPCs, like NEPCs, show disregard for androgen-targeted therapies. ADT has been shown to drive the trans-differentiation and focal NED of CRPC, known as therapy-induced NEPC (t-NEPC) [297]. When examining tissue samples from PRAD patients that have undergone ADT, focal populations of NEPCs were found in nearly all samples [284].

Additional studies are needed to better understand the role of chronic inflammation signaling in PCa castration resistance, stemness, NED, metabolic reprogramming, chemoresistance, metastasis, and immunosuppression. Based on what we know already and findings in other solid tumors, we can postulate that the presence of chronic inflammation in the prostate increases a patient’s risk of developing PCa and, if left uncontrolled, there may be an emergence of aggressive PCa phenotypes such as CRPCs, PCSCs, and NEPCs, leading to PCa progression (Figure 14).

## 11. Conclusions

The role of tumor heterogeneity in fueling PCa progression, recurrence, and aggressiveness has been extensively studied [298,299,300]. NED and stemness are early markers for cancer recurrence and progression. Although the maintenance and enrichment of NEPCs and PCSCs in prostate tumors have been associated with poor prognosis and limited treatment options, our knowledge of the oncogenic role of chronic inflammation signaling in these events is still embryonic [278]. During pro-tumor inflammation, several inflammatory molecules are either activated or suppressed for a prolonged time, resulting in undesirable consequences that encourage the growth and enrichment of aggressive tumor phenotypes in the tumor microenvironment. The contributions of many of these inflammatory signaling activators (TLRs, MYD88, TRIF, IL1R1, IRAK1/4, TRAF6, and NF-κB) and suppressors (IRAK3, TOLLIP, SOCS1, RNF216, IL1RL1, IL36RN, and TNFAIP3) in PCa remain poorly studied, despite the increasing association in the scientific literature of chronic inflammation with PCa initiation, progression, immunosuppression, and therapy resistance [301]. This limited mechanistic information is also peculiar to chronic prostatitis and BPH.

The recent discovery of the functional status of TLR/IL1R signaling in PCa has further rekindled interest in this area of research. The overexpression of TLR3, TLR4, and TLR9 in prostate biopsies and histological tissues is predictive and associated with biochemical recurrence in PCa patients [302], thus challenging the long-established view that these receptors are solely functional in innate immune cells. Similar observations have been found in the case of IRAKs and other downstream inflammatory molecules [205,209]. IRAK1 is overexpressed in many tumors and has been demonstrated to either directly or indirectly contribute to tumor progression and aggressiveness [221]. An improved understanding of the association between IRAK signaling and prostate tumorigenesis will provide a template for studies in other prostate conditions.

Another justification for targeting IRAKs is based on the notion that prostate cells have varied expression patterns of PAMP and DAMP receptors. Moreover, these receptors have non-specific ligandability. The lack of potent pan-PRR modifiers makes it difficult to target the aberrant activation of several PRRs simultaneously. In the same manner, the role of the activation of NF-κB transcription factors in PCa carcinogenesis has been extensively studied, but is still poorly defined. This is probably due to the systemic toxicity, selectivity, and bioavailability issues that have marred the development of potent, efficacious, and safe small molecule inhibitors to target dysregulated NF-κB expression for the clinical treatment of PCa. Moreover, the fact that NF-κB transcription factors can be activated by multiple (canonical and non-canonical) pathways further complicates and limits the clinical success of the available NF-κB inhibitors. The non-availability of an FDA-approved compound for NF-κB despite years of study justifies the need to explore alternative upstream and midstream targets and develop better chemical probes with clinical significance. Since IRAK1 signaling has been shown to activate the downstream NF-κB transcription factors in both immune and cancer cells, targeting IRAK1 in PCa may be an alternative and therapeutically feasible approach. Hence, we propose that targeting IRAK signaling may provide a better therapeutic or prophylactic strategy for the management of chronic inflammation-driven prostate pathologies.

To gain a better and more extensive insight into the association between chronic inflammation and prostate diseases, the following questions, among others, need to be answered:I.How do dysregulation and genetic alteration of each inflammatory molecule, including IL-1Rs, TLRs, TLR adapters (such as MyD88, Mal/TIRAP, CD14, TRAM, and TRIF), and midstream to downstream molecules (such as IRAKs, TRAF6, TAK1/2, TAB1/2/3, JNKs, ERK1/2, MAPK1-14, PI3K isoforms, AKTs, mTOR, NF-κB transcription factors), promote the development and progression of prostate diseases?II.What roles do inflammatory suppressors/repressors such as IRAK3, TOLLIP, SOCS1, RNF216, IL1RL1, IL36RN, and TNFAIP3 play in the pathogenesis of prostate diseases?III.Do IRAKs and other midstream inflammatory molecules have any clinical significance in the pathogenesis of prostate diseases?IV.Can inflammatory molecules be effectively used as diagnostic and prognostic biomarkers for predicting PCa progression, chronic prostatitis, and recurrent BPH?V.What is the oncogenic role of each IRAK family member in PCa chemoresistance, angiogenesis, NED, stemness, castration resistance, cancer proliferation, and metastasis, among others?VI.Which of the inflammatory genes play the biggest role in recurring prostatic disease conditions?VII.What are the best ways to therapeutically target aberrant and chronic inflammatory signaling in prostatic diseases?VIII.What is the role of each component of the Inflammasomes or NLR signaling in promoting the pathogenesis of prostate disease conditions?IX.What are the roles of the PI3K/AKT/mTOR pathway signaling in chronic inflammation-driven BPH and PCa?X.Does dysregulation or persistent activation of interferon signals affect prostate disease development, progression, or recurrence?XI.Does aberrant RIG-1-like receptor (RLR) signaling play any significant role in the persistence of prostatic disease conditions?XII.What roles do cGAS-STING signaling play in chronic prostatitis, BPH, and PCa?XIII.Are there microbial signatures to distinguish between prostatitis, BPH, and PCa?XIV.Does microbial dysbiosis contribute to racial disparities in prostate disease conditions?XV.Can microbial dysbiosis signatures be used as diagnostic and prognostic biomarkers for prostatic diseases?XVI.What are the molecular mechanisms linking prostatic diseases with other chronic inflammatory conditions, including inflammatory bowel disease (IBD) and diabetes mellitus (DM)?XVII.Are there novel strategies that can be explored to develop prophylactic or therapeutic vaccines against chronic prostatitis, BPH, and PCa?

Lastly, large-scale multi-racial integrative genomic, transcriptomic, epigenomic, proteomic, metabolomic, and microbiomic studies should be carried out to help in solving the missing pieces of the inflammatory jigsaw puzzle associated with chronic prostatitis, BPH, and PCa.

## Figures and Tables

**Figure 1 cancers-15-03110-f001:**
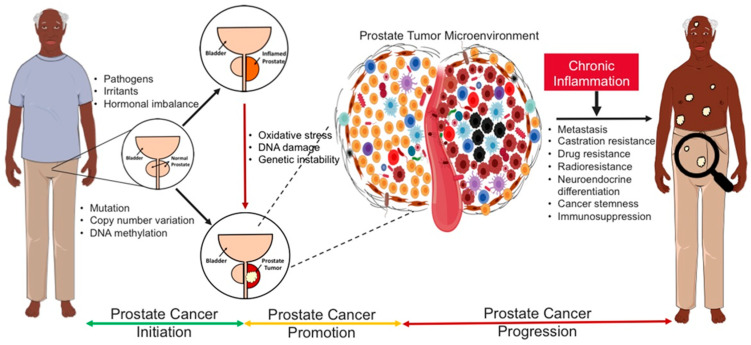
A diagrammatic illustration of the link between chronic inflammation and different stages of prostate tumorigenesis, including prostate cancer initiation, promotion, and progression.

**Figure 2 cancers-15-03110-f002:**
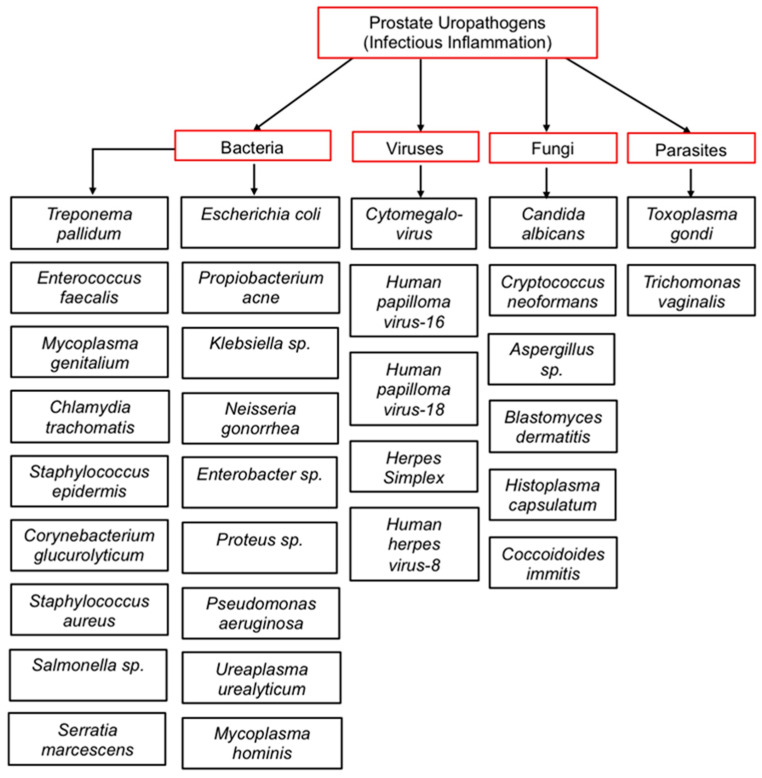
Flow chart highlighting some of the common uro-pathogens (bacteria, viruses, fungi, and parasites/protozoans of the prostate/urinary tract microbiota) that have been isolated from healthy and diseased prostate tissues using several molecular techniques.

**Figure 3 cancers-15-03110-f003:**
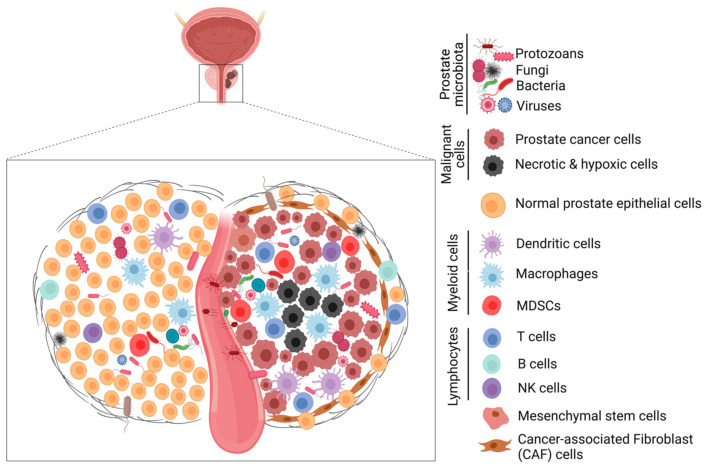
Diagrammatic illustration of the prostate tumor microenvironment showing the cell subpopulations and microbiota. Created with Biorender.com.

**Figure 4 cancers-15-03110-f004:**
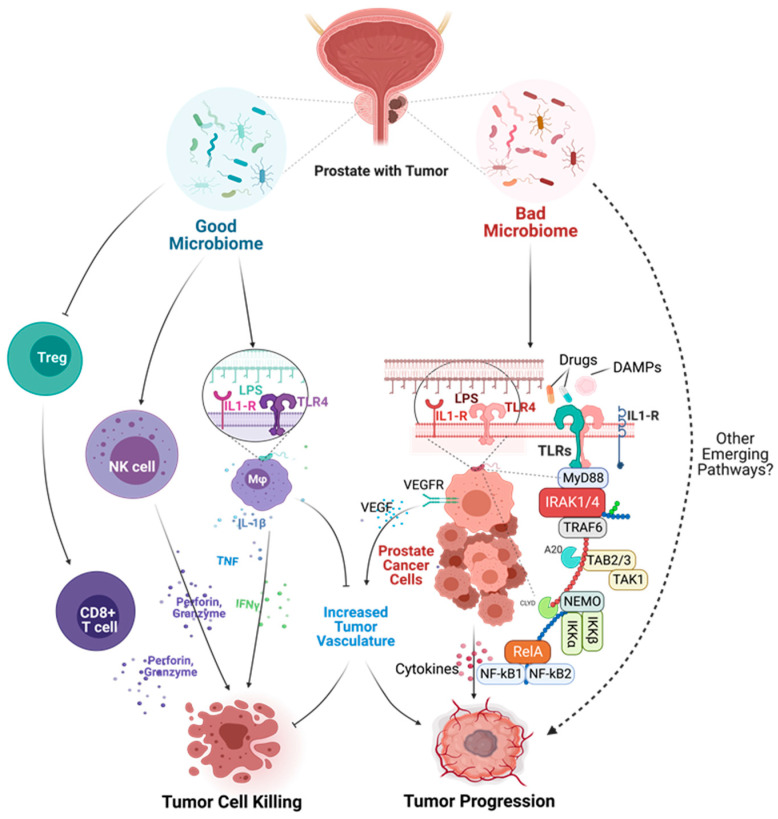
A schematic illustration of the prostate microbiome as it relates to the mechanisms involved in anti-tumor immune response versus tumor progression response in the prostate tumor microenvironment. Created with BioRender.com.

**Figure 5 cancers-15-03110-f005:**
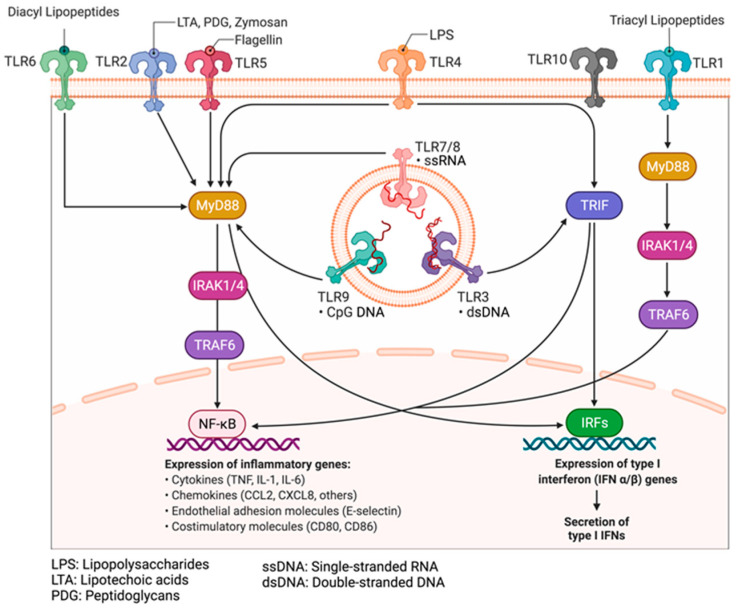
A schematic illustration of the activation process of TLR receptors and their downstream signaling molecules upon activation forming the TLR-MYD88-IRAK1/4-TRAF6-NF-κB/IRF or TLR3/TLR4-TRIF-IRF/NF-κB signaling pathways. The illustration also includes the location of common mammalian TLRs, either on the cell membrane or within the cell compartments (Endosome), and their corresponding ligands or agonists. Created with BioRender.com.

**Figure 6 cancers-15-03110-f006:**
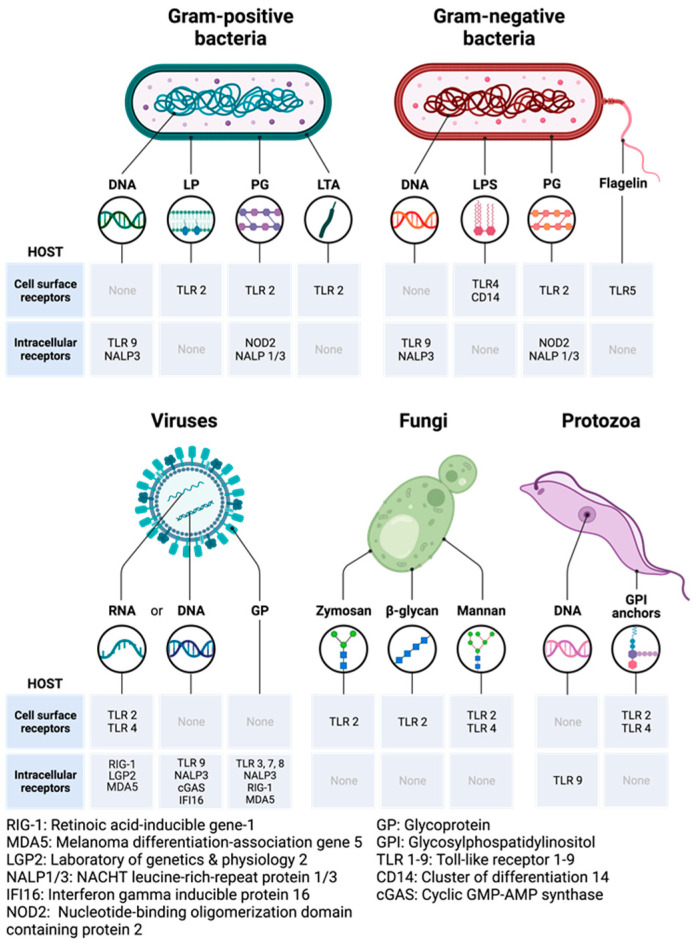
Diagrammatic illustration of common cellular and intracellular PRR receptors in bacteria, viruses, fungi, and protozoa. Created with Biorender.com.

**Figure 7 cancers-15-03110-f007:**
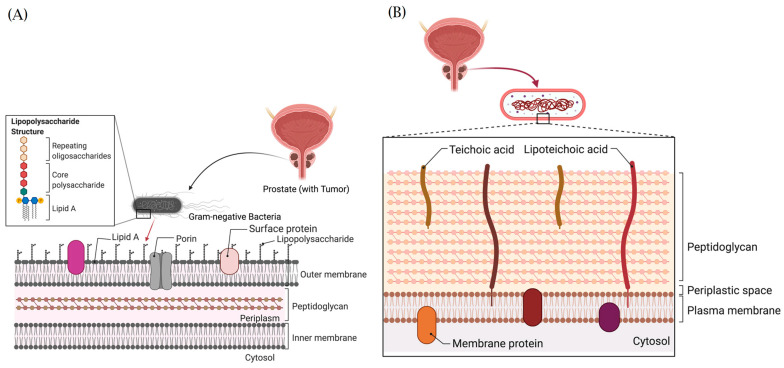
Schematic Illustrations of the cell wall components of Gram-negative (**A**) and Gram-positive (**B**) bacterial organisms. The LPS of Gram-negative bacteria is very important for danger signaling through the activation of TLR4, while the LTA of Gram-positive bacteria activates the TLR2 signaling pathway. Created with BioRender.com.

**Figure 8 cancers-15-03110-f008:**
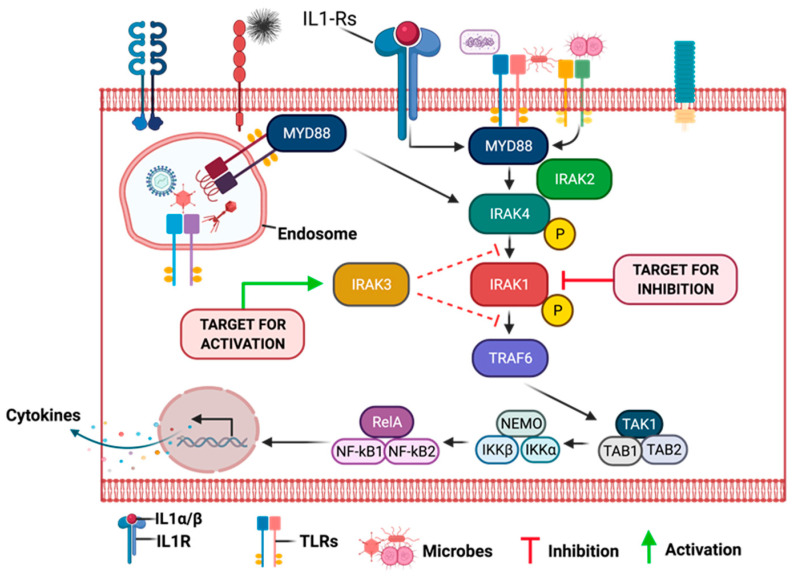
A schematic illustration of the TLR/IL1/IRAK/NF-κB inflammatory signaling pathway axis. The diagram also displays the potential novel therapeutic targets against inflammation-driven PCa progression. In our lab, we are interested in understanding the role of IRAKs in prostate tumorigenesis as well as evaluating the potential of targeting their signaling pathways. IL1Rs and TLRs are activated by their ligands and/or PAMPs/DAMPs, thus inducing a series of downstream inflammatory signaling cascades regulated by IRAKs at the midstream level. It is worth elucidating the cellular and molecular mechanisms involved in chronic inflammation-driven cancer growth, metastasis, chemoresistance, metabolic reprogramming, stemness, immune evasion, and neuroendocrine (NE) differentiation. Created with BioRender.com.

**Figure 9 cancers-15-03110-f009:**
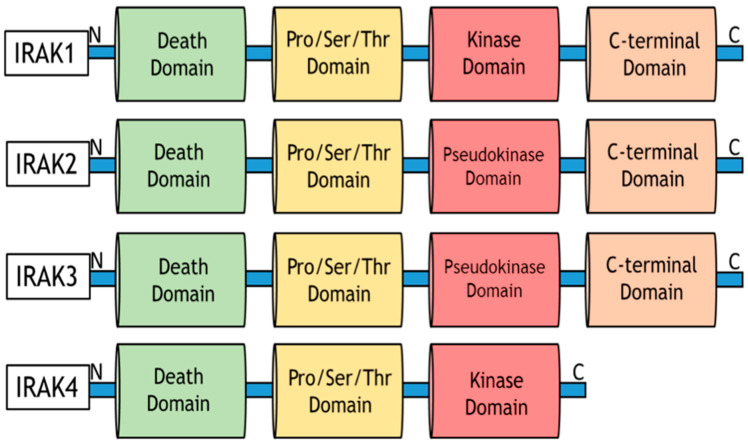
The structural architecture of the protein domains of IRAK1–4. The illustration shows the different domains important for binding (Death domain and C-terminal domain), catalytic activity (Kinase domain), and autophosphorylation (Kinase domain and Pro/Ser/Thr domain).

**Figure 10 cancers-15-03110-f010:**
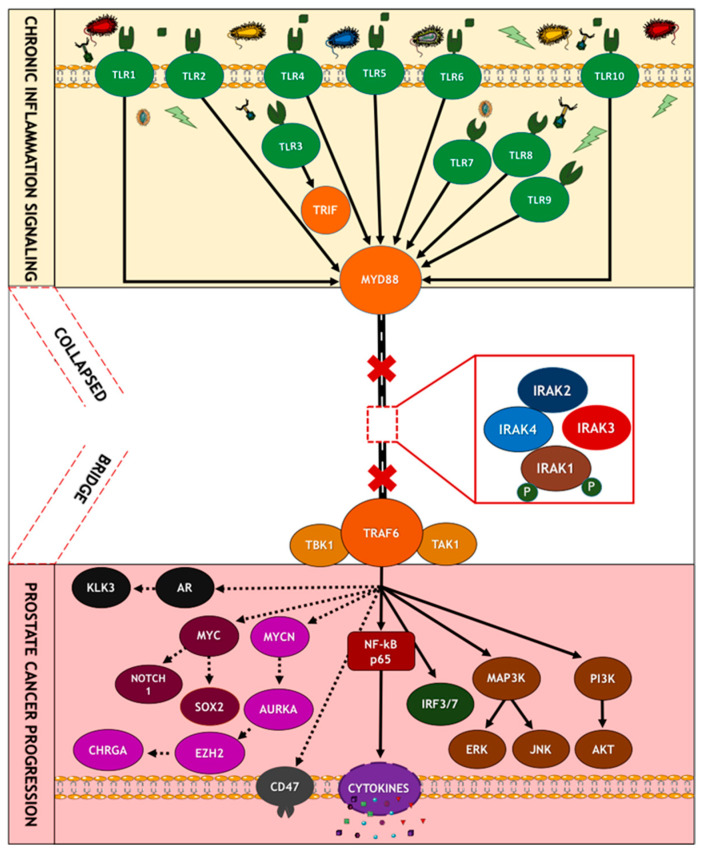
Diagrammatic illustration showing the possible crosstalk between genes in the TLR/IRAK/NF-κB signaling pathway and PCa-associated oncogenes/tumor suppressors.

**Figure 11 cancers-15-03110-f011:**
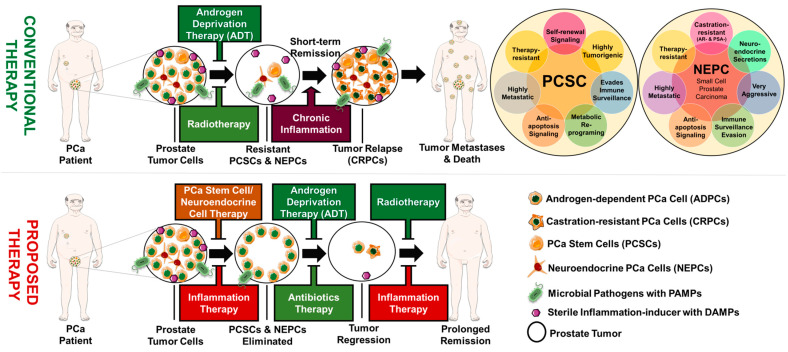
A schematic illustration of one of the possible novel therapeutic strategies that could be explored for the management of chronic inflammation-driven PCa, as envisaged in our lab.

**Figure 12 cancers-15-03110-f012:**
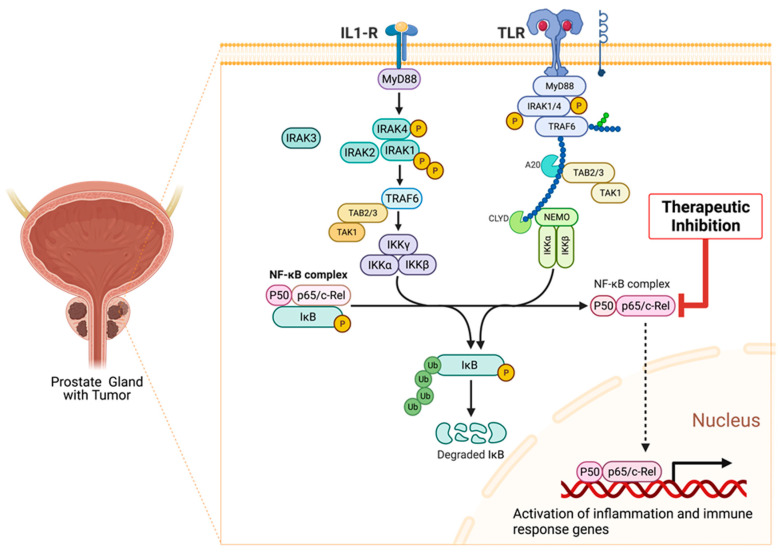
A schematic illustration of the IL1-R/TLR/NF-κB pathway with emphasis on activation of different NF-κB subunits via the IRAK1/TRAF6-mediated pathway. Created with BioRender.com.

**Figure 13 cancers-15-03110-f013:**
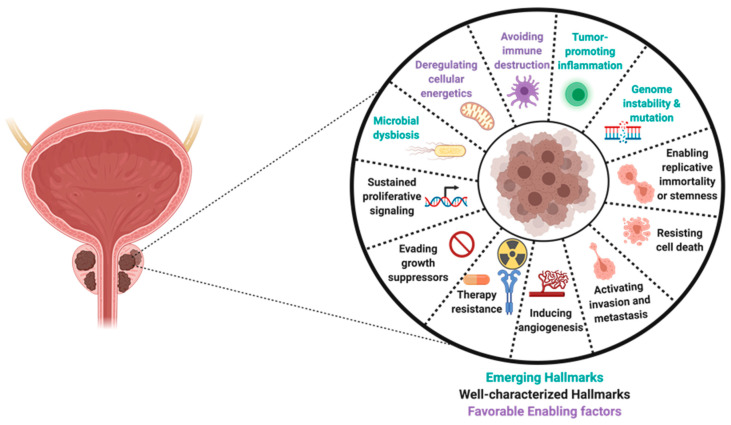
A schematical illustration of hallmarks of prostate cancer progression, including well-characterized hallmarks and the emerging hallmarks. Chronic or tumor-promoting inflammation, microbial dysbiosis, and genome instability are some of the emerging hallmarks. Created with BioRender.com.

**Figure 14 cancers-15-03110-f014:**
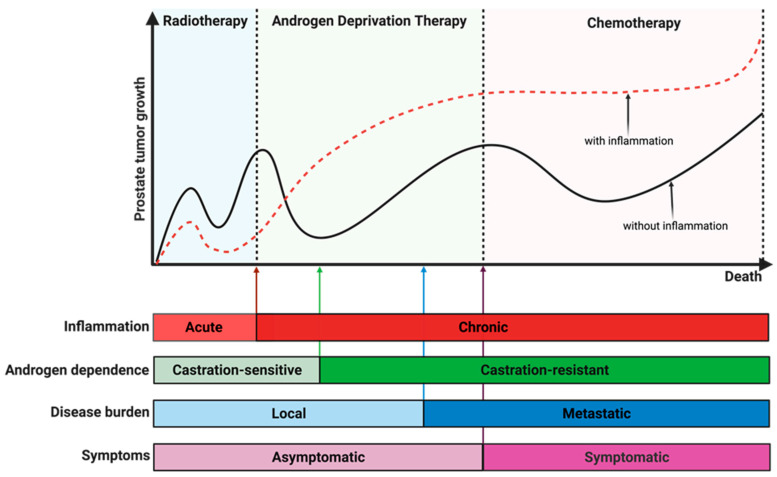
A schematic illustration of the aggressive evolution of prostate tumor progression from androgen-sensitive to castration-resistant and from local to metastatic in the presence or absence of chronic inflammation. A few studies have provided evidence that chronic inflammation can enhance chemoresistance, castration resistance, and radio-resistance. Created with BioRender.com.

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
