# Peer review of "The Molecular Basis and Clinical Consequences of Chronic Inflammation in Prostatic Diseases: Prostatitis, Benign Prostatic Hyperplasia, and Prostate Cancer"

_cancers, 2023, doi:10.3390/cancers15123110_

Round 1

Reviewer 1 Report

The Authors provide an interesting review on the role of chronic inflammation in prostatic diseases, including prostate cancer, a very common cancer worldwide.

The text is long and exhaustive and overall well written, it goes very deep into the molecular mechanisms that lead to chronic prostatitis and prostate cancer. There are 14 figures and more than 200 refs. In fact it looks more like a book chapter than a review.

Some paragraphs such as 6 on the role of the microbiome and 8 on racial disparities are interesting and original. Instead , paragraph 9 on the molecular mechanisms of chronic inflammation and carcinogenesis is particularly long. The Authors may consider to shorten it a bit. For instance, the part of TLR and NF-kB activation by PAMPs and DAMPs is described well, but is really long and I would perhaps restrict it to prostate studies. Table 1 is not needed.

The other problem I found is that many refs are obsolete, there are 6 from 2020, only 2 from 2021, none from 2022.  An update of references is necessary.

Author Response

Authors’ Response 1: We would like to thank the reviewer for their comments and feedback on our manuscript. We have given thorough consideration to your suggestions, particularly regarding section 9 titled "Mechanisms of oncogenic inflammatory signal transduction." In response, we have made significant improvements to this section by eliminating unnecessary jargon and condensing the paragraph as much as possible. Additionally, as requested, we have removed Table 1 from the manuscript.

Authors’ Response 2: We understand the importance of up-to-date references in scientific literature and appreciate the reviewer's feedback on this matter. As the reviewer pointed out, there were only two references from 2021 and none from 2022 in the original version of our paper. We have made an effort to include some recent publications (2021-2023) in our revised version while ensuring the manuscript remains as concise as possible. However, we would like to emphasize that this review paper/manuscript was originally completed in the spring of 2021, which explains the limited number of references beyond that timeframe. We also want to inform the reviewer that the delay in submitting the manuscript for review was due to the unfortunate passing of Dr. James Kumi-Diaka, one of the esteemed senior authors of this paper. Dr. Kumi-Diaka lost his battle with cancer last year, and as he was initially the corresponding author, it caused a significant delay in the submission process. We hope that the updated references will address the reviewer's concerns, and we appreciate their feedback in helping us to improve the quality of our paper. Thank you again for your time and attention.

Comments for Editors

During the revision process of this manuscript, we have addressed several key points raised by the reviewers. We made corrections to the citations and referencing style, removed unnecessary paragraphs, and followed the reviewers' suggestion to exclude Table 1. Additionally, we have diligently rectified grammatical and typographical errors that were identified during the revision. In accordance with the editors' request, we have also included the summary statement and graphical abstract.

Thank you for your patience, understanding, and guidance throughout this process.

Reviewer 2 Report

Very well written and comprehensive review, discussing almost all aspects of the role of chronic inflammation in prostate pathology.

In this reviewer`s opinion it is proper to include in the title that this is a narrative review

Author Response

Authors’ Response 1: Thank you very much for your positive feedback on our manuscript. We are glad to hear that you found our review paper to be well-written and comprehensive in its coverage of the molecular basis and clinical consequences of chronic inflammation in prostatic diseases. We appreciate your time and effort in reviewing our paper and providing us with valuable feedback.

Authors’ Response 2: We acknowledge the suggestion to include in the title that this is a narrative review. However, considering the specific guidelines and criteria provided by the journal during manuscript submission, we have opted to leave the decision regarding the title to the discretion of the editors. We appreciate your attention to this matter and thank you for bringing it to our attention.

Comments for Editors

During the revision process of this manuscript, we have addressed several key points raised by the reviewers. We made corrections to the citations and referencing style, removed unnecessary paragraphs, and followed the reviewers' suggestion to exclude Table 1. Additionally, we have diligently rectified grammatical and typographical errors that were identified during the revision. In accordance with the editors' request, we have also included the summary statement and graphical abstract.

Thank you for your patience, understanding, and guidance throughout this process.
